# Decreasing mitochondrial RNA polymerase activity reverses biased inheritance of hypersuppressive mtDNA

**Daniel Corbi**[1]*, **Angelika Amon**[1,2†]

**1** David H. Koch Institute for Integrative Cancer Research and the Department of Biology, Massachusetts Institute of Technology, Cambridge, Massachusetts, United States of America, **2** Howard Hughes Medical Institute, Massachusetts Institute of Technology, Cambridge, Massachusetts, United States of America

† Deceased.
* dan.corbi@gmail.com

**Data Availability Statement:** Most relevant data is contained within the manuscript; for a few datasets of raw RNA or DNA sequencing please see the following links: SMRT sequencing of Hypersuppressive mtDNAs: https://doi.org/10.

## Abstract

Faithful inheritance of mitochondrial DNA (mtDNA) is crucial for cellular respiration/oxidative phosphorylation and mitochondrial membrane potential. However, how mtDNA is transmitted to progeny is not fully understood. We utilized hypersuppressive mtDNA, a class of respiratory deficient *Saccharomyces cerevisiae* mtDNA that is preferentially inherited over wild-type mtDNA (*rho+*), to uncover the factors governing mtDNA inheritance. We found that some regions of *rho+* mtDNA persisted while others were lost after a specific hypersuppressive takeover indicating that hypersuppressive preferential inheritance may partially be due to active destruction of *rho+* mtDNA. From a multicopy suppression screen, we found that overexpression of putative mitochondrial RNA exonuclease *PET127* reduced biased inheritance of a subset of hypersuppressive genomes. This suppression required *PET127* binding to the mitochondrial RNA polymerase *RPO41* but not *PET127* exonuclease activity. A temperature-sensitive allele of *RPO41* improved *rho+* mtDNA inheritance over a specific hypersuppressive mtDNA at semi-permissive temperatures revealing a previously unknown role for *rho+* transcription in promoting hypersuppressive mtDNA inheritance.

## Author summary

Functional mitochondrial DNA is important for cells to maintain fitness and too much damaged mitochondrial DNA can cause debilitating diseases in humans. Inheritance of mitochondrial DNA from cell to cell as they divide is still a poorly understood process especially when multiple mitochondrial DNA forms are present. Here, we study a defective yeast mitochondrial DNA with the unusual property that it is inherited almost exclusively when present with functional mitochondrial DNA inside the same cell. We noticed that the functional mitochondrial DNA becomes damaged which suggests the defective mitochondrial DNA somehow promotes the destruction of the functional mitochondrial DNA when both are present; whereas, the current thinking is that the defective mitochondrial DNA is simply made faster than functional mitochondrial DNA. Also, we found that

5061/dryad.547d7wm8v RNA sequencing
characterizing nuclease dead Pet127 mutants:
https://doi.org/10.5061/dryad.cfxpnvx5z RNA
sequencing of overexpressed pet127 mutants:
https://doi.org/10.5061/dryad.mw6m905xg.

**Funding:** AA received an award from the National
Institute of General Medical Sciences (GM118066)
which helped fund this project. https://www.nigms.
nih.gov/ AA was an investigator of the Howard
Hughes Medical Institute which helped fund this
project. https://www.hhmi.org/ The funders had no
role in study design, data collection and analysis,
decision to publish, or preparation of the
manuscript.

**Competing interests:** The authors have declared
that no competing interests exist. Author Angelika
Amon was unable to confirm their authorship
contributions. On their behalf, the corresponding
author has reported their contributions to the best
of their knowledge.

a reduction of functional mitochondrial DNA gene expression protects the functional mitochondrial DNA from destruction by defective mitochondrial DNA revealing a novel role for functional mitochondrial DNA in the preferential inheritance of defective mitochondrial DNA. Both findings suggest there is an interplay between the genomes, either by competition for resources or interaction between the genomes, which has not been previously considered.

## Introduction

The mitochondrion is an essential eukaryotic organelle and the site for many critical metabolic reactions such as iron-sulfur cluster metabolism, heme biosynthesis, the TCA cycle, and cellular respiration/oxidative phosphorylation [1–3]. Most mitochondrial proteins are encoded by the nuclear DNA, made in the cytosol and imported into mitochondria [4,5]. However, mitochondria also have their own genome (mtDNA) containing a small number of genes that encode core components of complexes involved in cellular respiration/oxidative phosphorylation and machinery to translate those genes [6,7]. Unlike nuclear DNA, there are many copies of mtDNA per cell and, in *Saccharomyces cerevisiae*, both linear and circular forms of mtDNA are present [8–10].

In budding yeast, respiration-capable mtDNA (*rho+*) encodes genes critical for activity of the mitochondrial ribosome (*VAR1*, *15srRNA*, *21srRNA*, *tRNA*), components of mitochondrial electron transport chain (ETC) complexes: complex III (*COB*), complex IV (*COX1*, *COX2*, *COX3*), and components of the ATP synthase complex (*ATP6*, *ATP8*, *ATP9*) [6,7]. Loss of respiratory function caused either by partial disruption of mtDNA (*rho-*) or total loss of the mtDNA (*rho0*) results in impaired cellular respiration/oxidative phosphorylation and a slower colony growth phenotype. This reduced growth occurs even in non-respiratory conditions and is called petiteness [11,12]. Although mtDNA must be faithfully replicated, maintained, and segregated to progeny to maintain cellular fitness, mtDNA inheritance is not fully understood [13–17].

An approach to understand mtDNA inheritance comes from studies of mutant mtDNA in yeast that exhibit extremely biased inheritance. When two yeast cells mate, their cytoplasms fuse and each haploid parent contributes mitochondria with mtDNA to the resulting diploid daughter cell [10]. In most cases, mating between yeast parents containing *rho+* respiration competent mtDNA and parents containing respiration incompetent *rho-* or *rho0* mtDNA results in almost entirely respiration competent progeny, indicating strong inheritance of *rho+* mtDNA [18]. Some *rho-* genomes may persist in the progeny after mating *rho+* and *rho-* yeast, and these *rho-* genomes are called partial suppressives [18,19]. However, a subset of *rho-* mutants, called hypersuppressive (HS) mtDNA, are extremely biased in their inheritance. When mated with *rho+* cells, cells with *HS* mtDNA result in greater than 95% *rho-* progeny [18,20]. *HS* mtDNA mutants display preferential mtDNA inheritance despite giving rise to slow-growing respiration-incompetent cells. Thus, understanding how HS mutants hijack mtDNA inheritance machinery provides insights into how mtDNA inheritance works.

Characterization of *HS rho-* mutant genomes showed that they consist entirely of short (less than 2.5kbp) tandem repeats of one of three regions of high sequence similarity from the *rho+* genome [21–23]. As the HS genomes found in the diploid progeny after mating *rho+* and *HS rho-* are unchanged from those in the *HS rho-* parent, a prevailing theory is that HS mutants confer a replication advantage over *rho+* [19,22,24]. The regions of the *rho+* genome present in *HS* mutants are thought to be origins of replication, having similarities to mammalian

mtDNA replication origins [25–27], and are known as *ORI* or rep regions [23,28–30] despite not being necessary for mtDNA stability [31]. According to the replication advantage theory, *HS* mtDNAs have a higher density of *ORI* regions, because of the small tandem repeats, confering a replicative advantage over *rho+* mtDNA upon mating. Preferential inheritance of damaged mtDNA has been implicated in the progression of human mtDNA diseases and aging [32,33]. As there is similarity between HS origins in yeast and the heavy strand origin in mammals [26], there is reason to believe that preferential inheritance of mtDNA is similar. So, by understanding the principles of biased mtDNA inheritance in yeast we may gain insights into disease contexts.

One model supporting the replication advantage theory is the RNA priming hypothesis for mtDNA replication. The *rho+* genomic regions corresponding to the *HS ORIs* direct transcription of a ~300bp non-coding RNA that is cleaved and can be used as a primer for *in vitro* DNA replication [7,29,30]. The *rho+* genome has eight regions of *ORI* homology [34,35], but only the three or four *HS ORIs* (2, 3, 5 and, in some strain backgrounds, 1) are known to make such an RNA [7,27]. The presence of only RNA producing *ORIs* in HS mutants suggests that HS genomes are replicated by RNA priming initiation and this mechanism confers their replication advantage over *rho+* genomes [7].

There is, however, evidence against the RNA priming hypothesis for mtDNA replication. Certain *rho-* mtDNA genomes are stably replicated despite lacking either an *ORI* promoter or the mitochondrial RNA polymerase [31,36,37]. Also, HS mtDNA was shown to still be preferentially inherited without the mitochondrial RNA polymerase [38]; although, there are serious caveats to this experiment. Thus, both replication and HS biased inheritance need not work through *ORI* RNA or, more broadly, an RNA intermediate. A recombination-based replication model has been proposed to resolve this discrepancy [39,40].

There are some indications that the replication advantage model cannot entirely explain HS biased inheritance. The replication rate of a panel of partially suppressive *rho-* mutants fails to correlate with the extent of inheritance bias [41]. Also, increasing the pool of available nucleotides reduces the HS inheritance bias [42]. This suggests that alternative models of biased inheritance are partially or wholly responsible for the HS phenotype.

Here, we uncover that a specific *HS* mtDNA causes DNA damage to *rho+* mtDNA. Also, we perform a forward genetic suppressor screen using a high-copy genomic library to look for multicopy suppressors of HS. We find that overexpression of mitochondrial RNA exonuclease *PET127* suppresses the HS inheritance bias of certain HS alleles by negatively regulating mitochondrial RNA polymerase *RPO41*.

## Results

### Characterization of *HS* mtDNA

To generate HS mutants, we first used low-level ethidium bromide (EtBr) treatment for short periods of time to generate petite colonies (S1A Fig, [11]). Next, to identify which petite colonies were *HS rho-*, we mated the newly generated petite strains to *rho+* cells and monitored the color of the diploid strains. Monitoring color allowed us to take advantage of the fact that respiration is required for biosynthesis of a red purine analog that turns *ade2-1* yeast red [43]. Thus, while *rho+ ade2-1* yeast colonies turn red, *rho0* or *rho- ade2-1* colonies remain white [43]. By definition, HS cells will result predominantly in *rho-* diploids upon mating with *rho+*. Thus, we identified HS alleles by looking for white colonies upon mating petite colonies with *rho+* (S1B Fig). We validated the potential HS alleles using a quantitative mating assay where we mated the cells, plated the cells to select for diploids, and replica-plated the diploids to medium requiring respiration. We then can assess the fraction of mated cells which retain

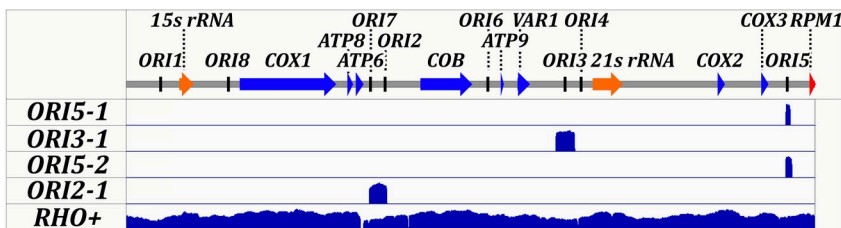

**Fig 1. Generation and Characterization of HS Alleles.** Mitochondria DNA from HS1b (*ORI5-1*), HS3b (*ORI5-2*), HS6a (*ORI3-1*), HS11b (*ORI2-1*), and *rho+* control cells was sequenced using pacific biosciences sequencing technology and mapped back to reference genome. Reads from *HS ORI5-1* mapped to base pairs 82,101 to 82,615 of the wild type reference genome. *HS ORI3-1* mapped to 53,495 to 55,836. *HS ORI5-2* mapped to 82,095 to 82,817. *HS ORI2-1* mapped to 30,305 to 32,497. Scale bar ranges are 0 to 1619 for *HS ORI5-1*, 0 to 1474 for *HS ORI5-1*, 0 to 1111 for *HS ORI3-1*, 0 to 660 for *HS ORI2-1* and 0 to 305 for *rho+*.

*rho+* mtDNA by dividing the number of cells which grow on respiration requiring medium by the total number of cells on the diploid selection (S1C Fig, [18]).

Four of the HS clones were further characterized. We performed sequencing using single molecule real-time (SMRT) sequencing of the mtDNA from each clone and the *rho+* parent and mapped the sequenced DNA back to the wild-type reference genome (Fig 1). In accordance with past literature, we found the four clones mapped to the mtDNA at *ORI* containing regions. We renamed the alleles *HS ORI5-1*, *HS ORI5-2* (spanning all of *HS ORI5-1*), *HS ORI3-1*, and *HS ORI2-1*. The long reads facilitated by SMRT sequencing allowed us to confirm that each of the HS alleles are tandem *ORI* repeats completely lacking other regions of the *rho+* genome.

## The presence of *HS ORI5-1* mtDNA damages *rho+* mtDNA

Previous studies showed that *HS* mtDNA eliminates *rho+* mtDNA, however it is unclear how the *rho+* mtDNA is eliminated [19,22]. To shed light on the fate of *rho+* mtDNA, we asked what happens to different regions of *rho+* mtDNA shortly after introducing *HS* mtDNA to cells. We mated *HS ORI5-1* and *rho+* parents and used PCR to ask if regions of *rho+* mtDNA are still present in the daughter colony. We found three types of respiration incompetent colonies. Five colonies had no observable *rho+* regions (Fig 2: Colonies 2, 3, 4, 9, 12), which is expected for the final state of cells after complete takeover of *HS* mtDNA [19,22]. Four colonies maintained *rho+* mtDNA from all regions assayed (Fig 2: Colonies 6, 7, 8, and 10) much like those from the *rho0* mated control and the *rho+* parent. This was unexpected as respiration incompetent colonies from a *rho+* and HS cross were thought to entirely contain HS genomic DNA [24]. Lastly, we observed three colonies that lost some but not all regions of the *rho+* genome (Fig 2: Colony 1 losing Cox1-Cterm; Colony 5 losing Cox2, Cox3, and Cox1-Cterm; and Colony 11 losing Cox2, Cox3, Atp9, and Cox1-Cterm). All colonies were assayed for a control nuclear genomic region. Only *rho0* parental controls failed to contain any mitochondrial DNA assayed by a nonspecific *ORI* primer set that detects the *ORI5* locus present in both *rho+* and *HS ORI5-1* mtDNA (Fig 2). The sensitivity of the *rho+* detecting primers was assayed by five-fold serial dilutions of *rho+* parental DNA into *HS ORI5-1* parental DNA (S2 Fig). The most sensitive primers are Cox1-Nterm, Cox3 and Atp9 which were all able to detect 0.64ng of *rho+* template. Whereas, Cox1-Cterm and Cox2 failed to detect 3.2 ng of parental DNA. For colony 1, a Cox2 band is present but not a Cox1-Cterminal band and as Cox2 is less sensitive than Cox1 C-terminal that indicates a loss of the Cox1 C-terminal region. For colony 5 and colony 11, Cox3, the most sensitive primer, is missing despite the presence of the Cox1-Nterm bands indicating a loss of Cox3. From these data we conclude that *rho+* mtDNA is lost in a

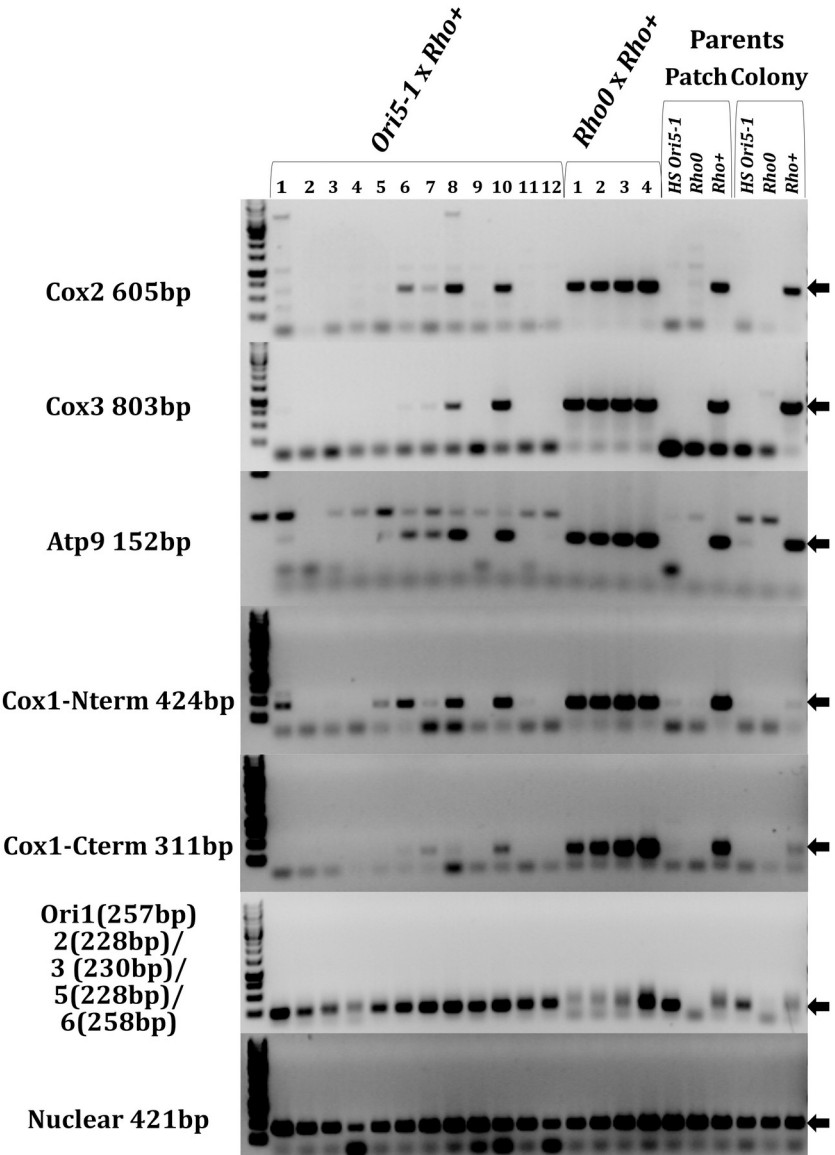

**Fig 2. Colony PCR after *HS* x *rho+* Reveals Remnants of *rho+* mtDNA.** The indicated strains were mated and plated on diploid selection medium. The parent strains were plated to single colonies. After two days, individual colonies were scraped off the plates, and DNA was isolated. Genomic DNA was normalized, and PCRs were performed with primer sets amplifying the indicated loci. Reactions were run on agarose gels and stained with ethidium bromide. Arrows indicate the size of the intended product.

piecemeal fashion rather than all at once. This fragmented loss of mtDNA cannot be explained by a replicative advantage, therefore *HS* mtDNA somehow causes genomic damage to the intact mtDNA.

## *PET127* overexpression suppresses the biased inheritance of a subset of HS genomes

To better understand the factors governing mtDNA inheritance, we performed a whole-genome screen to identify suppressors of the biased inheritance of HS mutants. We utilized the *HS ORI5-1* allele for our suppression screen as many prior studies of *HS* mtDNA have

focused on *ORI5* alleles [39,40,42,44]. Specifically, wild-type *rho+* cells were transformed with a high-copy plasmid library [45], mated with the *HS ORI5-1* strain, and assayed using the same red/white colony phenotype that we used to isolate HS alleles (Fig 3A) [43]. Most diploid colonies were white due to the dominant inheritance of the *HS ORI5-1* mutant. However, four plasmids were identified that repeatedly led to the formation of red *rho+* colonies, indicating suppression of the *HS* biased inheritance. Two colonies contained plasmids bearing a region of chromosome XV with four shared genes: the 3' region of *RTS1*, putative gene *YOR015W*, *ERP4*, and *PET127* (one plasmid fully encompassing *PET127*, one containing the 5' 2254 bp out of the 2403 bp coding sequence of *PET127*) (S3A Fig).

We mapped the gene required for suppression on the plasmids. The region of *RTS1* in the plasmids does not contain a functional promoter, and *YOR015W* is only a putative gene with a start site 45 bp from the end of *RTS1*, making them less likely candidates. Thus, we focused on *ERP4* and *PET127* and quantitatively assessed their individual suppressive capabilities (Fig 3B). An empty vector in the *rho+* background showed 1.7% ± 1.3 *rho+* colonies when mated to *HS ORI5-1* but yeast containing the parent plasmid isolated from the screen recovered 42% ± 2.1 *rho+* colonies. A plasmid containing either *ERP4* or *PET127* showed 3.5% ± 3.2 and 43% ± 5.4 *rho+* colonies respectively, implicating *PET127* as the sole causative suppressive gene. The *rho+* colonies resulting from high-copy *PET127* suppression grew in a subset region of the original diploid parental colony (S3B Fig). This *rho+* subset was not sectored in any discernable pattern, and, as the high-copy plasmid is selected for on the diploid selection plates and should not be lost in large sections of the colony, indicates that there is stochasticity in how high-copy *PET127* prevents HS takeover. Mating *rho+* yeast containing the suppressive plasmid or a plasmid containing *PET127* to the *rho0* control did not result in a reduction of *rho+* colonies. This indicated that the high-copy *PET127* plasmid expression level from its endogenous promoter did not result in the loss of mtDNA in the *rho+* haploid parent despite prior literature showing expression of *PET127* from the strong *ADC1* promoter on multicopy plasmids can result in petite cells [46]. From these data, we conclude that overexpression of *PET127* is a suppressor of the preferential inheritance of *HS ORI5-1*.

Next, we asked if overexpression of *PET127* could suppress the phenotype of the other *HS rho-* mitochondrial genomes using the same approach. We mated *rho+* strains containing *PET127* with the one clone from each strain in the panel of HS alleles in S1C Fig and determined the number of *rho+* diploid colonies (Fig 3C). The suppressive effect of *PET127* overexpression was strongest for *HS ORI5-1* but also significant in *HS2b*, *HS5b*, *HS8b*, and *HS10b*. The effect on all other HS alleles was insignificant including *HS ORI5-2* which contains the whole region of *HS ORI5-1*. We genotyped HS alleles using a PCR strategy with primers which have homology to *ORI2*, *ORI3*, and *ORI5* but have 3' ends facing each other rather than 5' ends [47]. As HS alleles are repetitive in nature, this strategy should generate a product spanning the length of a repeat except for the nucleotides between the 3' primer ends. This strategy did not produce products for all HS alleles, but determined that the *PET127* sensitive strain *HS10b* contains *ORI2* DNA indicating that *PET127* sensitivity is not due to the features of one particular ORI.

We wondered whether either the specific HS allele or a background nuclear mutation is responsible for determining which strains are sensitive to high-copy *PET127* suppression. To determine which hypothesis is correct, we shuttled the mtDNA of *HS ORI5-1* and *HS ORI5-2* alleles to the same independently generated *rho0* recipient by cytoduction (mating with yeast defective in nuclear fusion to allow for transfer of cytoplasmic factors without changing nuclear genotype) and tested the suppression by quantitatively mating the resulting strains to *rho+* cells overexpressing *PET127* [48]. This experiment resulted in similar levels of *rho+* colonies post-mating (S3C Fig), indicating that differences in the extent of high-copy *PET127*

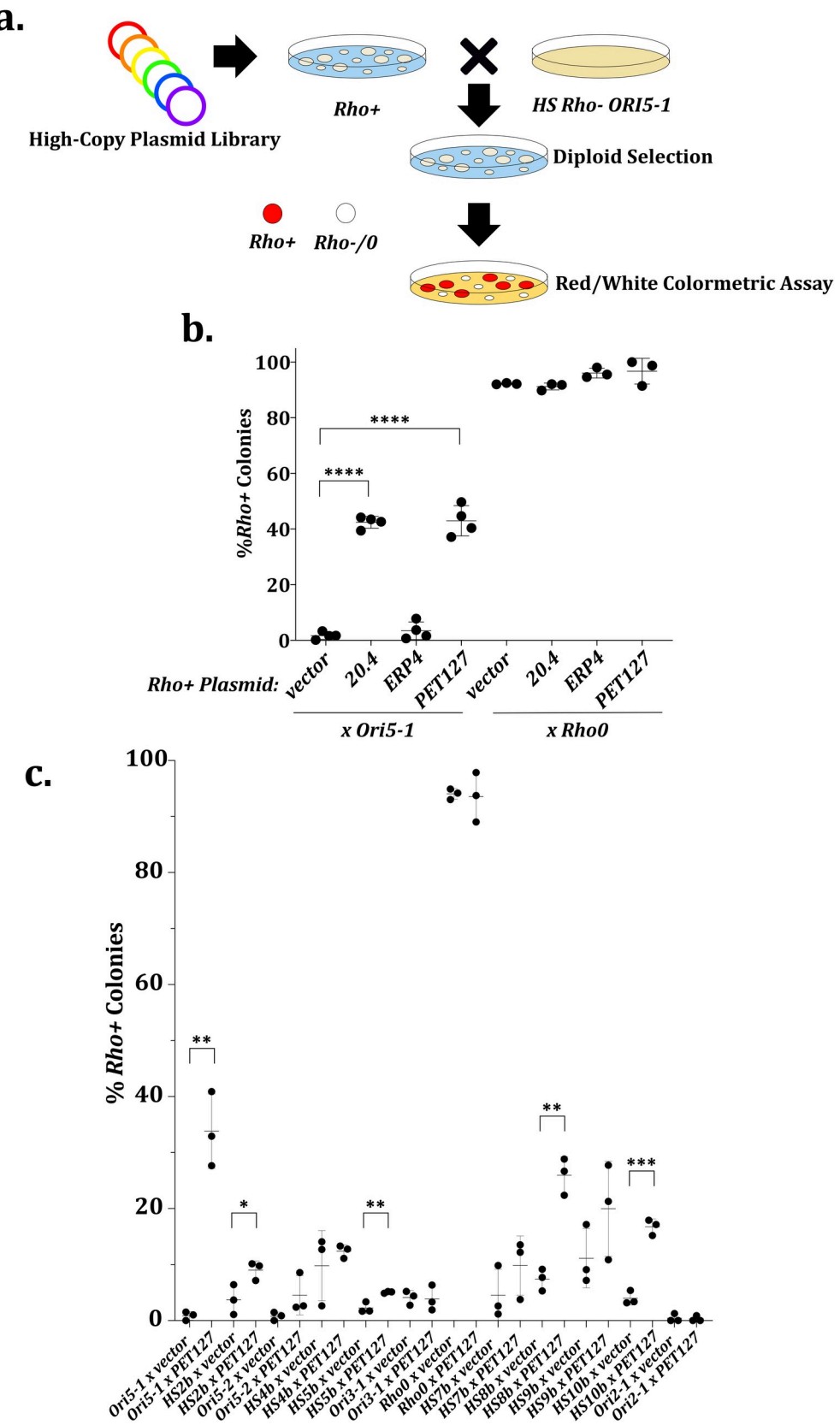

**Fig 3. High-copy *PET127* is a Suppressor of *HS rho-* Preferential Inheritance.** A. *Rho+* cells were transformed with Yep13 high-copy library [45] and 10,967 colonies were plated. Plasmid containing colonies were mated with lawns of *HS Ori5-1*, replica plated to diploid selection medium and replica plated again to YEPD with no additional adenine. We identified 154 red colonies initially, however only 5 colonies were red after repetition of the assay. Four of the five colonies showed suppression and were sanger sequenced using Yep13 sequencing primers. B. Quantitative mtDNA inheritance assay validating plasmid 20.4, identified by the screen, and the two candidate genes carried by 20.4. Significance was determined using one-way ANOVA separately on the *HS* (N = 4) and *rho0* (N = 3) crosses with means compared to the vector control and using Dunnett's multiple comparison's test. **** indicates adjusted P value less than 0.0001. All other comparisons were not significantly different. C. Quantitative mtDNA inheritance assay as in b testing high-copy *PET127* on the other isolated HS alleles. Significance was determined by student's unpaired T test on the *vector* and high-copy *PET127* pair for each *HS* or *rho0* set (N = 3). * indicates a P value between less than 0.05. ** indicates a P value less than 0.01. *** indicates a P value between 0.001. All other comparisons were not significantly different.

suppression are due to features of the mtDNA, not the nuclear DNA of the cells where the *HS* genomes were isolated. From these data we conclude that high-copy *PET127* is a suppressor of a subset of HS genomes, and that *PET127* sensitivity depends on the genome but can work on *ORI5* or *ORI2* based HS alleles.

## The Pet127 association with Rpo41 is responsible for the suppression of *HS ORI5-1*

To gain insights into the molecular mechanism by which high *PET127* levels suppress the preferential inheritance of *HS* mtDNA, we investigated which Pet127 function contributes to this phenotype. *PET127* is a nuclear-encoded mitochondrial gene that encodes a putative RNA exonuclease that localizes to the mitochondrial inner membrane [46]. *PET127* was discovered as a loss-of-function suppressor of C-terminal *PET122* mutations which block translation of *COX3* mRNA [49], and, while *PET127* deletions fail to cause the petite phenotype implied by the "PET" nomenclature [49], expression of *PET127* from the strong *ADC1* promoter on high-copy plasmids causes loss of mtDNA [46]. *PET127* is homologous to the PD-(D/E)XK exonuclease superfamily [50], and has been proposed to trim the 5' region of mitochondrial mRNAs prior to translation by the mitochondrial ribosome [46,51]. Thus, we first examined whether the 5' RNA exonuclease activity of *PET127* is required for suppression of the HS phenotype by mutating four conserved exonuclease active site residues to generate a nuclease-dead allele (*pet127-nd*) [46,49–52]. Deletion of *PET127* increases the size of mitochondrial transcripts, presumably by eliminating the nuclease activity required to trim the transcripts to the normal size [46,52]. To confirm that *pet127-nd* lacks nuclease activity, we performed RNA sequencing and looked for the accumulation of transcripts in regions where the *PET127* wild-type does not usually accumulate (S4 Fig). The RNA sequencing of cells harboring *pet127-nd* showed that the levels of mitochondrial RNA in genic regions was relatively unchanged with respect to *PET127* (S4A Fig), but the intergenic regions and non-coding RNAs showed accumulation of RNA that mimics that of *pet127Δ* (S4B and S4C Fig). These data are consistent with the *pet127-nd* mutant eliminating nuclease activity. Importantly, the *pet127-nd* mutant retained full suppression of HS preferential inheritance, as revealed by quantitative mating assay (40% ± 6.7 *rho+* colonies) (S5 Fig). Thus, excess RNA exonuclease activity from overexpression of *PET127* does not affect *HS ORI5-1* mtDNA inheritance.

Pet127 also interacts with the sole mitochondrial RNA polymerase, Rpo41, although its role in transcription is unclear [53]. Therefore, we tested whether the association of Pet127 with Rpo41 is required for the observed suppression. First, we verified the association between Pet127 and Rpo41 by co-immunoprecipitation. As specific antibodies are not available, we created epitope-tagged versions of *PET127* (*PET127-HA*) and *RPO41* (*V5-RPO41*).

Immunoprecipitation of V5-Rpo41 resulted in co-precipitation of Pet127-HA, indicating that these factors interact *in vivo* (Fig 4A).

To identify a *PET127* allele defective in *RPO41* association, we reconstituted the association *in vitro* using recombinant proteins expressed in bacteria. We expressed full length Pet127 lacking its predicted mitochondrial targeting sequence (aa 1–47) and GST-Rpo41 lacking its mitochondrial targeting sequence and performed a pulldown assay in mixed bacterial lysates [53,54]. The mitochondrial targeting sequences are cleaved in the mitochondria and excluded from the mature forms of Pet127 and Rpo41 in yeast. 6xHis-Pet127$^{48-800}$ co-precipitated with GST-Rpo41 but not GST alone, indicating a specific and direct interaction between the two proteins (Fig 4B). We then constructed a series of truncation alleles of *PET127* from both the N-terminus and the C-terminus (S6 Fig). The smallest *PET127* allele that bound Rpo41 contained amino acids 48–215 and the largest allele that failed to bind Rpo41 contained amino acids 213–800 (Fig 4B). Thus, amino acids 48–215 of Pet127 are sufficient for binding to Rpo41.

To test whether *PET127* truncations function similarly in yeast, we expressed two of the truncation alleles (including the mitochondrial targeting sequence): *pet127$^{Δ48-215}$*, which lacks the binding region, and *pet127$^{1-215}$* which contains the minimal Rpo41 binding region (Fig 4C). The expression level of both truncated proteins was lower than full-length Pet127 (Fig 4D, Inputs); however, both truncated proteins were imported into mitochondria as they were resistant to proteinase K treatment of a mitochondrial enriched preparation (S7 Fig). Yeast expressing the *pet127$^{1-215}$* allele exhibit a second larger species possibly indicative of uncleaved mitochondrial targeting sequence (Fig 4D). Consistent with this hypothesis, the larger band abundance is reduced by proteinase K degradation (S7 Fig). Co-immunoprecipitation showed that Rpo41 binding with Pet127-nd, the allele lacking the RNA exonuclease activity, was unchanged from wild-type and binding with Pet127$^{Δ48-215}$ was less efficient than with Pet127$^{1-215}$ (Fig 4D and 4E). Thus, the *PET127* truncation mutants are suitable, in yeast, to interrogate whether Rpo41 binding activity is responsible for high-copy *PET127* suppression of *HS ORI5-1* biased inheritance.

We next used quantitative mating of *HS ORI5-1* and *rho+* yeast carrying multi-copy plasmids with the truncation alleles to test their ability to suppress HS biased inheritance. In spite its relatively low basal protein abundance, overexpression of the *PET127* allele containing the small *RPO41* binding region (*pet127$^{1-215}$*) was sufficient to suppress HS inheritance (23% ± 9.8 *rho+* colonies versus 48% ± 0.44 *rho+* colonies for full-length *PET127*) (Fig 4F). Notably, elimination of the *RPO41* binding region completely abolished *PET127* suppression, as demonstrated by overexpressing the *pet127$^{Δ48-215}$* truncation (1.4% ± 0.73 *rho+* colonies). The suppressive capability of *pet127$^{1-215}$* overexpression is consistent with the observation that plasmid 30.2 isolated from the high copy suppression screen did not contain the full C-terminal region of *PET127* (S3A Fig). These data show that the region of *PET127* that binds to *RPO41* is both necessary and sufficient to suppress HS biased inheritance.

## Reduced mitochondrial transcription suppresses *HS Ori5-1*

We postulated that Pet127 binding to Rpo41 alters the RNA polymerase activity of Rpo41 and that altered mitochondrial RNA polymerase activity affects HS biased inheritance. To test this hypothesis, we asked whether increasing *RPO41* abundance to restore the stoichiometric ratio between Pet127 and Rpo41 amplifies or counteracts the suppressive effect of *PET127* overexpression on HS biased inheritance. We assessed the HS biased inheritance on *rho+* yeast carrying both a high-copy plasmid expressing *V5-RPO41* and a second high-copy plasmid expressing *PET127*. Quantitative mating with *HS ORI5-1* showed that the high-copy *PET127*

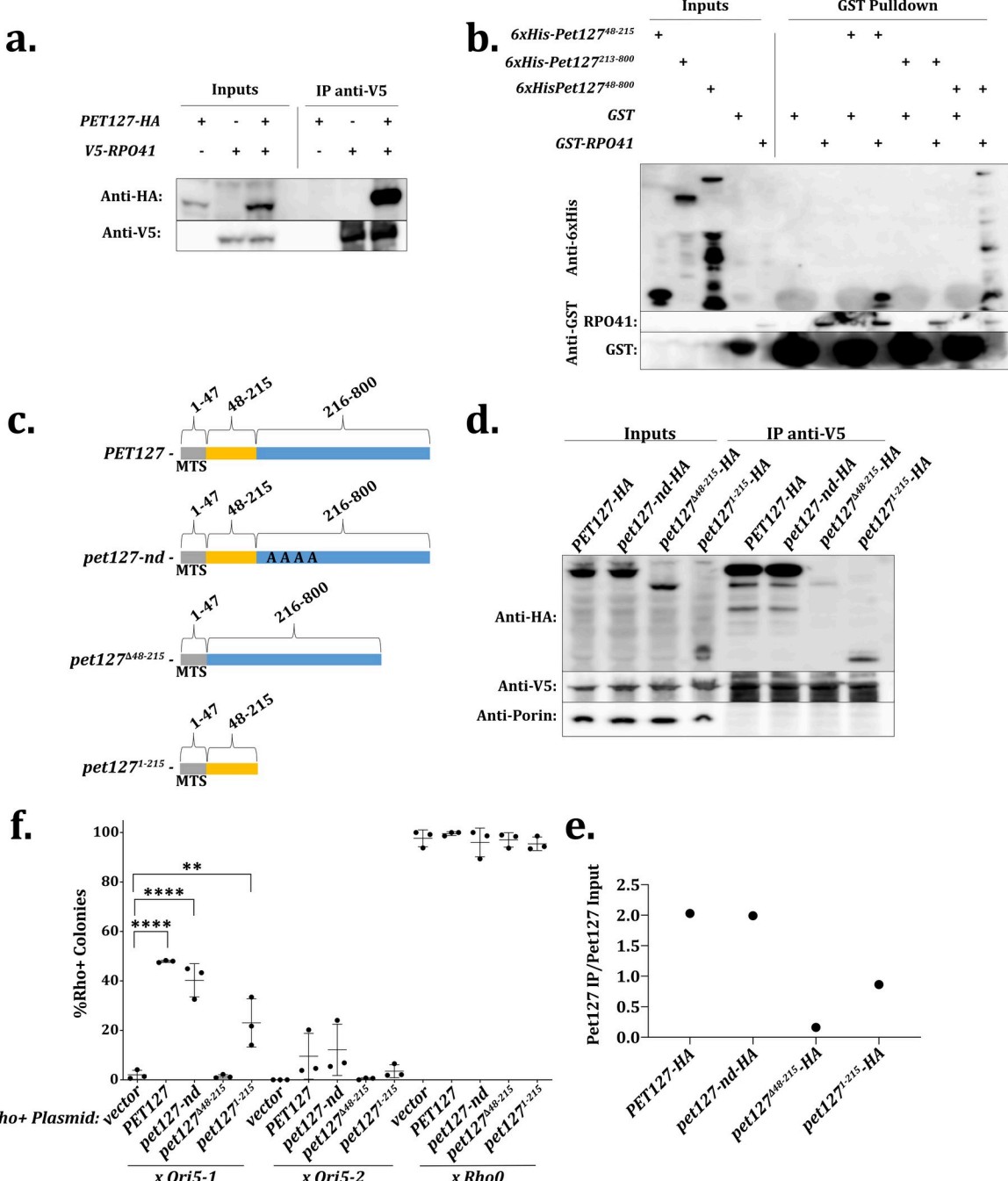

**Fig 4. *PET127* Binds to *RPO41* and the Binding Region is Responsible for Suppression of HS Biased Inheritance.** A. Coimmunoprecipitation of Rpo41 and Pet127. Cells expressing either *PET127-HA*, *V5-RPO41*, or *PET127-HA* and *V5-RPO41* were collected, lysed, and V5-Rpo41 was immunoprecipitated using anti-V5 conjugated beads. Samples were immunoblotted as indicated. B. Bacterial expression and lysate mixing of recombinant *RPO41* and *PET127* alleles. The indicated alleles were expressed in BL21 by the addition of IPTG. Cells were collected and lysed in a French press. For IPs, equal amounts of lysate from Pet127 construct expressing cells was mixed with lysate from either GST-Rpo41 or GST only expressing cells and precipitated with glutathione agarose beads. Samples were immunoblotted as indicated. C. Diagram of *PET127* alleles. The predicted mitochondrial targeting sequence is amino acids 1–47 in gray [54]. Amino acids 48–215 contain the Rpo41 binding region in yellow, and amino acids 216–800 contain the region lacking Rpo41 binding activity in blue. *pet127-nd* is a nuclease dead allele of *PET127* with amino acids predicted to be conserved active site residues changed to alanines: E346, D378, D391, and K393 [50]. D. Co-immunoprecipitation of *PET127-HA* alleles and *V5-RPO41* as in a. Samples were immunoblotted as indicated. Anti-porin used as a negative pulldown control. E. Quantitation of IP enrichment in d. F. Quantitative mtDNA inheritance assay with *rho+* high copy *PET127* alleles x *HS ORI5-1*, *HS ORI5*-2, or *rho0*. Significance was determined using one-

way ANOVA separately on each *HS* and *rho0* cross with means compared to the vector control using Dunnett's multiple comparison's test (N = 3). **** indicates adjusted P-value less than 0.0001 and ** indicates an adjusted P-value of 0.0025. All other comparisons were not significantly different.

plasmid decreased the inheritance of *HS* mtDNA (49% ± 7.9 *rho+* colonies) relative to control (4.2% ± 2.1 *rho+* colonies) (Fig 5A). In contrast, cells with the high-copy *V5-RPO41* plasmid showed no significant decrease in *rho+* colonies (0.55% ± 0.62 *rho+* colonies) relative to control cells (Fig 5A). Interestingly, cells with both *V5-RPO41* and *PET127* high-copy plasmids significantly abolished the suppression effect of the *PET127* plasmid alone (6.6% ± 0.94 *rho+* colonies). This finding is consistent with the hypothesis that excess Pet127 binding reduces the polymerase activity of Rpo41.The percent of rho+ colonies after mating *rho+* strains carrying *V5-RPO41* plasmid or both *V5-RPO41* and *PET127* plasmids to *rho0* cells was not significantly different (82% ± 10 in *V5-RPO41*, 88% ± 6.6 in *PET127 V5-RPO41*) from the strains not carrying the *V5-RPO41* plasmid (96% ± 2.4 in strain carrying both empty vectors, 97% ± 1.0 in *PET127*) (Fig 5A). From these data, we conclude that *RPO41* and *PET127* act in opposition to one another, and, because increasing Pet127 affects HS biased inheritance through Rpo41, there is a strong possibility that Pet127 is an inhibitor of Rpo41 polymerase activity.

As Rpo41 is an RNA polymerase, an increase in Rpo41 abundance should increase mitochondrial transcription, and, if *PET127* is an inhibitor of *RPO41*, increased Pet127 abundance should reduce mitochondrial transcription. Evaluating the mRNA levels of the mitochondrial coding genes *COX3*, *COX2*, and *ATP9* via RT-qPCR showed that, for all loci assessed, the addition of high-copy *V5-RPO41* increases mRNA relative to the control (Figs 5B and S8A). Co-expression of high-copy *PET127* and high-copy *V5-RPO41* greatly reduced the effect of *RPO41*, supporting an inhibitory role for Pet127 in transcription (Figs 5B and S8A). Expression of high-copy *PET127* alone showed a decreasing trend in mRNA levels, with significant effect on *COX3* mRNA, but insignificant effects on *ATP9* and *COX2* (Figs 5B and S8A). None of the tested conditions affected mtDNA levels (S8B Fig).

To further examine the inhibitory role of Pet127, we maximized its effect on transcription by replacing the wild-type promoter on the high-copy *PET127* alleles with the strong galactose-inducible *GAL1/10* promoter. In haploid *rho+* cells carrying the *pGal-PET127* plasmid three hours after galactose addition, *COX3* RNA levels were cut in half by qPCR compared to control strains while mtDNA levels were unaffected (S9A Fig). RNA sequencing performed five hours after galactose addition showed a similar trend for all major mitochondrial gene transcripts relative to the time of galactose addition (S9B Fig). Overall, these data indicate that high level of *PET127* expression causes a general reduction of mitochondrial transcription, supporting the hypothesis that Pet127 is an inhibitor of Rpo41.

In the RNA priming model for mtDNA replication, transcription from the mtDNA *ORI* regions is hypothesized to play a role in DNA replication, raising the possibility that the *PET127* overexpression effect on the HS phenotype is a consequence of altered *ORI* transcription. We therefore monitored *ORI* RNA, utilizing qPCR primers which amplify a transcript, which corresponds to a common area in all "transcription active" *ORI*s (2, 3, 5, ~230bp) as well as a hypothetical-long transcript, common to *ORI1 and ORI6* (~250bp). Surprisingly, *ORI* RNA levels were increased upon *pGAL-PET127* expression in the *rho+* background and the *HS ORI5-1* background (S9C and S9D Fig). However, this increase was independent of Pet127 binding to Rpo41 as the *pGal-pet127*$^{Δ48-215}$ truncation mutant had the same effect as wild-type *PET127* (S9C Fig). Expression of the nuclease dead mutant caused *ORI* RNA to increase above that of the *pGal-PET127* allele which suggests that the nuclease activity affects the abundance of *ORI* transcripts. From these data, we conclude that the levels of the 230 or 250 bp-long *ORI*

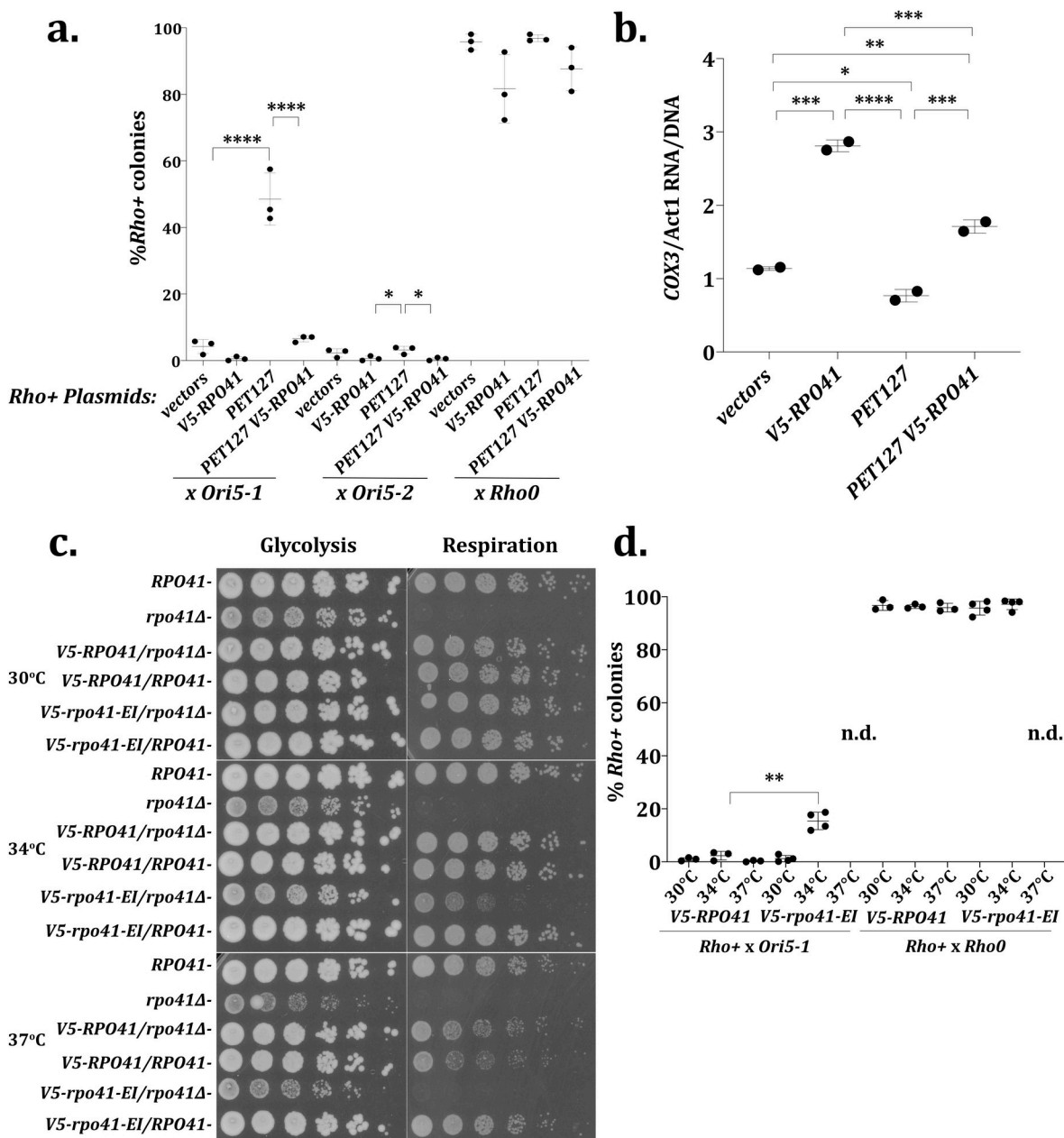

**Fig 5. Reduction of Transcription from the Mitochondrial RNA pol, *RPO41*, Suppresses HS Biased Inheritance.** A. Quantitative mtDNA inheritance assay on strains carrying all combinations of high-copy *PET127*, high-copy *V5-RPO41*, and their respective empty vector controls. Significance was determined using one-way ANOVA separately on each *HS* and *rho0* cross with means compared to all other samples and using Tukey's multiple comparison's test (N = 3). **** indicates adjusted P value less than 0.0001. * indicates adjusted P-value less than 0.05. All other comparisons were not significantly different. B. RT-qPCR of *COX3*/*ACT1* RNA divided by qPCR of *COX3*/*ACT1* DNA on the high-copy *PET127* high-copy *V5-RPO41* combination strains collected at high cell density to increase respiration. Significance was determined using one-way ANOVA comparing all values with each other using Tukey's multiple comparison's test (N = 2). **** indicates adjusted P-value less than 0.0001, *** indicates an adjusted P-value of 0.0008 to 0.0001, ** indicates an adjusted P-value of 0.0057, * indicates an adjusted P-value of 0.0275. C. Five-fold serial dilution of strains containing *RPO41* alleles beginning at 1.1 x 10$^7$ cells/ml plated on glycolysis (YEPD) or respiration (YEPG) utilizing media. Plates were incubated at either 30˚C or 34˚C for 2 days or 37˚C for 3 days. D. Quantitative mtDNA inheritance assay of *rho+ V5-RPO41* x either *HS ORI5-1* or *rho0 V5-RPO41* and *rho+ v5-rpo41-EI* x either *HS ORI5-1* or *rho0 v5-rpo41-EI* at 30˚C, 34˚C, and 37˚C. *v5-rpo41-EI* cross at 37˚C was not determined as there was synthetic lethality on the diploid selection plates (see S11 Fig). Significance was determined using student's T-test on each temperature between *V5-RPO41* and *v5-rpo41-EI* in *HS* (N = 4) and *rho0* (N = 3) cross. ** indicates a P-value of 0.0015. All other comparisons were not significantly different.

transcripts do not play a role in *PET127* suppression, but that *PET127* expression and nuclease activity can affect levels of those transcripts.

As Pet127 is an inhibitor of Rpo41 when overexpressed, we wanted to know if there are conditions where Pet127 levels change relative to Rpo41 and their inhibitory relationship might be important for cellular life. When *Saccharomyces cerevisiae* undergo fermentation, a fermentable carbon source such as glucose gets converted into the respiratory carbon source ethanol [55]. In batch cultures, the fermentable carbon source is used up as cell density increases [56]. *Saccharomyces cerevisiae* can then utilize the ethanol as a carbon source, and this transition is called diauxic shift [57]. As Rpo41 activity is necessary for respiration, we decided to track Pet127 and Rpo41 protein levels as cells undergo diauxic shift. We found that as cell density increases and diauxic shift occurs Pet127 levels decrease relative to Rpo41 (S10A Fig). Conversely, diluting high density cultures into glucose medium increases Pet127 levels relative to Rpo41 (S10A Fig). We wondered whether the observed decrease was specific to ethanol conditions or whether other respiratory carbon sources cause the same effect. To test this, we grew cells in glucose and then split the culture four ways into medium containing either glucose or the respiratory carbon sources glycerol, acetate, or ethanol and waited to let the cells acclimate to the new conditions. We found that while Rpo41 levels remained the same, Pet127 protein levels decreased in all respiratory conditions (S10B Fig). From these data, we conclude that Pet127 protein levels decrease relative to Rpo41 in respiratory conditions.

To directly assess whether biased inheritance of HS can be prevented by inhibiting Rpo41-mediated mitochondrial transcription we attempted to reduce but not eliminate Rpo41 activity in the course of *rho+* and *HS* mating. Since *rho+* mtDNA is unstable in the *rpo41Δ* background [58], we utilized a temperature sensitive allele of *RPO41*, *v5-rpo41-EI* [53], at a semi-permissive temperature to assess HS preferential inheritance with a partially functional RNA polymerase. We tested the temperature sensitivity of *v5-rpo41-EI* at 30˚C, 34˚C, and 37˚C by testing the viability of serial dilutions of exponentially growing yeast in conditions either requiring respiration for growth or allowing for growth using glycolysis (Fig 5C). The *v5-rpo41-EI* allele permitted haploid cells to grow normally in respiration conditions at 30˚C, similar to wild type *RPO41*, but eliminated growth at 37˚C, mimicking *rpo41Δ*, and caused an intermediate reduction of growth at 34˚C (Fig 5C).

Quantitative mating of *HS ORI5-1* and *rho+* at 37˚C, in the *V5-RPO41* background showed no changes relative to the other temperatures (0.31% ± 0.29 *rho+* colonies), but the *v5-rpo41-EI* background showed a reduction of diploid colonies on synthetic dropout plates at 37˚C despite being a non-respiratory growth condition (S11 Fig). Because the result would be distorted by the synthetic lethal effect present on the diploid selection plate, we did not assess the percent of *rho+* colonies in the *v5-rpo41-EI* at 37˚C when mated with either *HS ORI5-1* or *rho0*. However, quantitative mating of *HS ORI5-1* and *rho+* at 34˚C resulted in a small but significant increase in *rho+* colonies in the *v5-rpo41-EI* background (15% ± 3.3) over the *V5-RPO41* background (2.3% ± 1.7 *rho+* colonies) (Fig 5D). No change was observed when mating *rho+* and *HS ORI5-1* at 30˚C in either the *V5-RPO41* background (1.0% ± 0.59) or the *v5-rpo41-EI* background (1.2% ± 1.2). As partial reduction of mitochondrial RNA polymerase activity is sufficient to suppress the preferential inheritance of *HS* mtDNA, we conclude that transcription of mitochondrial RNA promotes the *HS ORI5-1* preferential inheritance over *rho+* mtDNA.

## Discussion

### *HS ORI5-1* mtDNA damages *rho+* mtDNA

The main hypothesis regarding the mechanism of HS biased inheritance is that the HS mtDNA is better replicated and outcompetes the *rho+* mtDNA. However, we show here that

during *HS ORI5-1* takeover mtDNA of some regions of *rho+* mtDNA are eliminated while other regions remain. If purely a replication competition was responsible for the HS takeover, a uniform loss of the entire *rho+* genome would be expected rather than loss of some regions before others. Therefore, the introduction of *HS ORI5-1* mtDNA actively eliminates regions of the *rho+* genome. The idea that HS biased inheritance is not exclusively due to replicative advantage is consistent with the observation that there is no correlation between replication rates of different partially suppressive *rho-* mtDNAs and the extent to which they take over the *rho+* mtDNA [41].

How does *HS ORI5-1* mtDNA cause selective elimination of *rho+* mtDNA? HS mtDNA could cause a defect in *rho+* replication. Replication defects that could cause selective genome loss include a subset of origins failing or elongation defects preventing complete genome replication. Alternatively, *HS* mtDNA could directly recombine with *rho+* mtDNA with unequal crossovers causing destruction of *rho+* regions. A recombination DNA destruction mechanism provides an alternative explanation, besides recombination-based replicative-advantage, as to why recombination mutants are reported to reduce HS biased inheritance [39,40].

The HS phenotype damaging *rho+* mtDNA is a harmonious solution to some of the problems of the replicative-advantage model. The replicative advantage model has problems accounting for how there are so few *rho+* cell progeny from a HS cross. The replication advantage model proposes that the replication rate of HS genomes is so superior that almost no progeny obtain *rho+* mtDNA. The *rho+* containing parents then die due to replicative aging. It is known that *rho+* mtDNA is present in the progeny of the first division of a zygote [44]. Once it is inherited to cells, the *rho+* mtDNA would quickly become a pure population [59]. However, according to the DNA damage model, *rho+* mtDNA would have DNA damage which allows it to appear in progeny without a respiration phenotype.

## Pet127 is a negative regulator of Rpo41

Although, we show that Pet127 binds and represses Rpo41 function, the molecular mechanism is still unclear. Pet127 binding could be blocking a region required for Rpo41 binding with DNA. Alternatively, Pet127 binding could cause a conformational change in Rpo41 rendering it unable to transcribe RNA or bind to DNA. It is unclear how cells utilize the inhibitory relationship of Pet127 with Rpo41. Pet127 inhibition of Rpo41 could have a buffering role on Rpo41 to allow the cell to respond to fluctuations of Rpo41 levels. Rpo41 activity may be required more in respiratory conditions and Pet127 protein levels dropping in respiratory conditions may be a way cells can regulate Rpo41 activity.

## Biased inheritance of *ORI5-1* hypersuppressive mtDNA utilizes mitochondrial RNA polymerase activity

We show that transcription is involved in the HS phenotype because modulation of mitochondrial RNA polymerase activity alters the phenotype. Although a role for transcription is clear, the exact mechanism of transcription involvement in HS is not. The theory that mitochondrial RNA is a required intermediate in HS biased inheritance was previously tested by assessing HS biased inheritance in the *rpo41Δ* background lacking mtRNA [38]. Since *rho+* mtDNA is unstable in the *rpo41Δ* background, a non-preferentially inherited *"neutral" rho-* mtDNA was used in place of *rho+*. These studies showed that HS mtDNA was preferentially inherited over *neutral rho-* mtDNA in *rpo41Δ* cells. The problem with this experiment is that *neutral rho-* mtDNA is not an appropriate substitute for *rho+* mtDNA, as *rho+* mtDNA is inherited far better and has a different transcriptional landscape than the *neutral rho-* mtDNA. By utilizing the temperature sensitive *v5-rpo41-EI* allele at semi-permissive temperatures, we were able to

maintain the stability of *rho+* mtDNA while reducing mitochondrial transcription and show that mitochondrial RNA polymerase, *RPO41*, plays a role in HS biased inheritance.

How mitochondrial RNA plays a role in HS biased inheritance remains unclear. Prior thinking is that the only transcript in HS alleles is *ORI* RNA, so transcription must be involved in the HS phenotype through production of *ORI* RNA. However, we show here that changes in *ORI* RNA levels do not correlate with changes in HS phenotype across *PET127* alleles. Although this finding does not completely rule out a role for *ORI* RNA in the HS phenotype, it is unlikely that *PET127* overexpression reverses HS biased inheritance through *ORI* RNA effects. Instead, we show that reduction of the HS phenotype occurs in concert with a general reduction of *rho+* transcription. How *rho+* transcription could be involved in the HS phenotype has not previously been considered. It is possible that a specific *rho+* transcript or gene product promotes HS biased inheritance. Alternatively, general *rho+* transcription or translation could be required. Given the lack of effects on the *ORI* transcript, it is unclear which specific transcripts are candidates and how, mechanistically, they would impact inheritance.

There are several possible models for how general transcription reduction could reduce biased inheritance. One possibility is that transcription machinery physically blocks *rho+* replication on the same stretch of DNA. If so, then removing RNA polymerase from the genome allows replication machinery access. While it has been shown that transcription can block nuclear replication initiation [60] and replication fork progression [61], it is unclear which, if either, method is responsible for preventing *rho+* replication.

Alternatively, a general reduction of transcription could increase the pool of available ribonucleotides. Hyperactivating RNR pathway, which increases nucleotide availability, partially suppresses HS biased inheritance, indicating that nucleotides are limiting for *rho+* mtDNA during HS takeover [42]. It is thought that ribonucleotides can substitute for deoxyribonucleotides in the yeast mtDNA during replication because of a promiscuous mtDNA polymerase and an absence of a mitochondrial ribonucleotide excision repair pathway [62]. Thus, reducing *rho+* mitochondrial mRNAs could increase the local ribonucleotide pool allowing for elongation utilizing ribonucleotides in a deoxyribonucleotide limited state.

It is puzzling that the methods of HS suppression described here only work on certain HS alleles and not on others. The implication is that the different alleles use different mechanisms for biased inheritance. it could be that the base repeat length of the HS allele determines the method of biased inheritance or that spontaneous mutations in the repeat sequence change the way HS alleles behave.

## Materials and methods

### Yeast strains and culture conditions

All strains are derivatives of W303 (AA2587) unless noted otherwise. See Table 1 for details. Liquid cultures were grown in YEPD(1% yeast extract, 2% peptone, 2% glucose) with additional adenine (55μg/L), uracil (22.4μg/L), and tryptophan (80μg/L) or SD media [2% glucose, 6.7 g/L yeast nitrogen base w/o amino acids (Difco), 2 g/L complete supplement mixture (without Histidine, without Leucine or without Uracil and Leucine, CSM MP Biomedicals)], to maintain plasmid selection, overnight at room temp (~23˚C for *rho+* strains) or 30˚C (for *rho-/0* strains) and diluted back to 3.3 x $10^6$ cells/ml and grown at 30˚C.

For galactose induction, strains were cultured overnight in SD-Histidine medium (*rho+* strains) or YEP medium (HS strains) with 2% raffinose instead of 2% glucose. Cells were diluted back to 3.3 x $10^6$ cells/ml in the same medium, allowed to grow one doubling, and the expression of the galactose promoter was induced by the addition of 1% galactose.

**Table 1. List of Yeast Strains Used in this Study.**

| Strain Number | Relevant Genotype | Fig |
|---|---|---|
| AA36030 | *MATalpha, ade2-1, leu2-3, ura3, trp1-1, HIS3+, can1-100 rho+* | S1a |
| AA41772 | *MATalpha, ade2-1, leu2-3, ura3, trp1-1, HIS3+, can1-100 HS Rho- **HS1a*** | S1c |
| AA41773 | *MATalpha, ade2-1, leu2-3, ura3, trp1-1, HIS3+, can1-100 **HS1b HS ORI5-1*** | **1, 2, 3b, 3c, 4f, 5a, S1c, S2, S3b S5, S9d** |
| AA41774 | *MATalpha, ade2-1, leu2-3, ura3, trp1-1, HIS3+, can1-100 HS Rho- **HS1c*** | S1c |
| AA41775 | *MATalpha, ade2-1, leu2-3, ura3, trp1-1, HIS3+, can1-100 HS Rho- **HS2a*** | S1c |
| AA41776 | *MATalpha, ade2-1, leu2-3, ura3, trp1-1, HIS3+, can1-100 HS Rho- **HS2b*** | 3c, S1c |
| AA41777 | *MATalpha, ade2-1, leu2-3, ura3, trp1-1, HIS3+, can1-100 HS Rho- **HS2c*** | S1c |
| AA41778 | *MATalpha, ade2-1, leu2-3, ura3, trp1-1, HIS3+, can1-100 HS Rho- **HS3a*** | S1c |
| AA41779 | *MATalpha, ade2-1, leu2-3, ura3, trp1-1, HIS3+, can1-100 **HS3b HS ORI5-2*** | **1, 3c, 4f, 5a, S1c, S3b, S5, S9d** |
| AA41780 | *MATalpha, ade2-1, leu2-3, ura3, trp1-1, HIS3+, can1-100 HS Rho- **HS3c*** | S1c |
| AA41781 | *MATalpha, ade2-1, leu2-3, ura3, trp1-1, HIS3+, can1-100 HS Rho- **HS4a*** | S1c |
| AA41782 | *MATalpha, ade2-1, leu2-3, ura3, trp1-1, HIS3+, can1-100 HS Rho- **HS4b*** | 3c, S1c |
| AA41783 | *MATalpha, ade2-1, leu2-3, ura3, trp1-1, HIS3+, can1-100 HS Rho- **HS4c*** | S1c |
| AA41784 | *MATalpha, ade2-1, leu2-3, ura3, trp1-1, HIS3+, can1-100 HS Rho- **HS5a*** | S1c |
| AA41785 | *MATalpha, ade2-1, leu2-3, ura3, trp1-1, HIS3+, can1-100 HS Rho- **HS5b*** | 3c, S1c |
| AA41786 | *MATalpha, ade2-1, leu2-3, ura3, trp1-1, HIS3+, can1-100 HS Rho- **HS5c*** | S1c |
| AA41787 | *MATalpha, ade2-1, leu2-3, ura3, trp1-1, HIS3+, can1-100 **HS6a HS ORI3-1*** | **1, 3c, S1c** |
| AA41788 | *MATalpha, ade2-1, leu2-3, ura3, trp1-1, HIS3+, can1-100 HS Rho- **HS6b*** | S1c |
| AA41789 | *MATalpha, ade2-1, leu2-3, ura3, trp1-1, HIS3+, can1-100 **HS6c rho0*** | **2, 3b, 3c, 4f, 5a, S1c, S3b S5** |
| AA41790 | *MATalpha, ade2-1, leu2-3, ura3, trp1-1, HIS3+, can1-100 HS Rho- **HS7a*** | S1c |
| AA41791 | *MATalpha, ade2-1, leu2-3, ura3, trp1-1, HIS3+, can1-100 HS Rho- **HS7b*** | 3c, S1c |
| AA41792 | *MATalpha, ade2-1, leu2-3, ura3, trp1-1, HIS3+, can1-100 HS Rho- **HS7c*** | S1c |
| AA41793 | *MATalpha, ade2-1, leu2-3, ura3, trp1-1, HIS3+, can1-100 HS Rho- **HS8a*** | S1c |
| AA41794 | *MATalpha, ade2-1, leu2-3, ura3, trp1-1, HIS3+, can1-100 HS Rho- **HS8b*** | 3c, S1c |
| AA41795 | *MATalpha, ade2-1, leu2-3, ura3, trp1-1, HIS3+, can1-100 HS Rho- **HS8c*** | S1c |
| AA41796 | *MATalpha, ade2-1, leu2-3, ura3, trp1-1, HIS3+, can1-100 HS Rho- **HS9a*** | S1c |
| AA41797 | *MATalpha, ade2-1, leu2-3, ura3, trp1-1, HIS3+, can1-100 HS Rho- **HS9b*** | 3c, S1c |
| AA41798 | *MATalpha, ade2-1, leu2-3, ura3, trp1-1, HIS3+, can1-100 HS Rho- **HS9c*** | S1c |
| AA41799 | *MATalpha, ade2-1, leu2-3, ura3, trp1-1, HIS3+, can1-100 HS Rho- **HS10a*** | S1c |
| AA41800 | *MATalpha, ade2-1, leu2-3, ura3, trp1-1, HIS3+, can1-100 HS Rho- **HS10b*** | 3c, S1c |
| AA41801 | *MATalpha, ade2-1, leu2-3, ura3, trp1-1, HIS3+, can1-100 HS Rho- **HS10c*** | S1c |
| AA41802 | *MATalpha, ade2-1, leu2-3, ura3, trp1-1, HIS3+, can1-100 HS Rho- **HS11a*** | S1c |
| AA41803 | *MATalpha, ade2-1, leu2-3, ura3, trp1-1, HIS3+, can1-100 **HS11b HS ORI2-1*** | **1, 3c, S1c** |
| AA41804 | *MATalpha, ade2-1, leu2-3, ura3, trp1-1, HIS3+, can1-100 HS Rho- **HS11c*** | S1c |
| AA41805 | *MATa, ade2-1, leu2-3, ura3, trp1-1, his3-11,15, can1-100, YEP13-20.4* | S3a |
| AA41806 | *MATa, ade2-1, leu2-3, ura3, trp1-1, his3-11,15, can1-100, YEP13-30.2* | S3a |
| AA41807 | *MATa, ade2-1, leu2-3, ura3, trp1-1, his3-11,15, can1-100, YEP13-39.1* | S3a |
| AA41808 | *MATa, ade2-1, leu2-3, ura3, trp1-1, his3-11,15, can1-100, YEP13-39.4* | S3a |
| AA39745 | *MATa, ade2-1, leu2-3, ura3, trp1-1, his3-11,15, can1-100, YEP13* | **2, 3b, 3c, 4f, S2, S3b, S5** |

*(Continued)*

**Table 1.** (Continued)

| Strain Number | Relevant Genotype | Fig |
|---|---|---|
| AA41322 | MATa, ade2-1, leu2-3, ura3, trp1-1, his3-11,15, can1-100, YEP13-20.4b (subclone of AA41805) | **3b** |
| AA39746 | MATa, ade2-1, leu2-3, ura3, trp1-1, his3-11,15, can1-100, YEP13-PET127 | **3b, 3c, 4f, S3b** |
| AA41323 | MATa, ade2-1, leu2-3, ura3, trp1-1, his3-11,15, can1-100, YEP13-ERP4 | **3b** |
| AA39257 | MATalpha, ura3-52, lys2-801, ade2-101, his3Del200, trp1Del63, leu2Del1, cyh2, kar1Del15, rho0 | **Materials and Methods** |
| AA39508 | MATa, ade2-1, leu2-3, ura3, trp1-1, HIS3+, can1-100, HS ORI5-1 rho- | **S3c** |
| AA40153 | MATa, ade2-1, leu2-3, ura3, trp1-1, HIS3+, can1-100, HS ORI5-2 rho- | **S3c** |
| AA38266 | MATa, ade2-1, leu2-3, ura3, trp1-1, HIS3+, can1-100, rho0 | **S3c** |
| AA39535 | MATalpha, ade2-1, leu2-3, ura3, trp1-1, his3-11,15, can1-100, YEP13 | **S3c** |
| AA39534 | MATalpha, ade2-1, leu2-3, ura3, trp1-1, his3-11,15, can1-100, YEP13-PET127 | **S3c** |
| AA39500 | MATa, ade2-1, leu2-3, ura3, trp1-1, his3-11,15, can1-100, PET127-6xHA::KANmx6 | **4a** |
| AA39479 | MATa, ade2-1, leu2-3, trp1-1, his3-11,15, can1-100, ura3::RPO41preseq-3xV5-RPO41::URA3, rpo41::KANmx6 | **4a** |
| AA39607 | MATa, ade2-1, leu2-3, trp1-1, his3-11,15, can1-100, ura3::RPO41preseq-3xV5-RPO41::URA3, rpo41::KANmx6, PET127-6xHA::KANmx6 | **4a, 4d, 4e, S10a, S10b** |
| AA2587 | MATa, ade2-1, leu2-3, ura3, trp1-1, his3-11,15, can1-100 | **1, S4a, S4b, S4c** |
| AA39672 | MATa, ade2-1, ura3, trp1-1, his3-11,15, can1-100, LEU2::pet127-nd | **S4a, S4b, S4c** |
| AA39670 | MATa, ade2-1, ura3, trp1-1, his3-11,15, can1-100, pet127::KANmx6, LEU2::pet127-nd | **S4a, S4b, S4c** |
| AA38088 | MATa, ade2-1, leu2-3, ura3, trp1-1, his3-11,15, can1-100, pet127::KANmx6 | **S4a, S4b, S4c** |
| AA40696 | MATa, ade2-1, leu2-3, trp1-1, his3-11,15, can1-100, pet127-nd-6xHA::HIS3, ura3::RPO41preseq-3xV5-RPO41::URA3, rpo41::KANmx6 | **4d, 4e** |
| AA40684 | MATa, ade2-1, leu2-3, trp1-1, his3-11,15, can1-100, ura3::RPO41preseq-3xV5-RPO41::URA3, rpo41::KANmx6, pet127(1–215)-6xHA::HIS3 | **4d, 4e** |
| AA40694 | MATa, ade2-1, leu2-3, trp1-1, his3-11,15, can1-100, pet127(Δ48–215)-6xHA::HIS3, ura3::RPO41preseq-3xV5-RPO41::URA3, rpo41::KANmx6 | **4d, 4e** |
| AA40287 | MATa, ade2-1, leu2-3, trp1-1, his3-11,15, can1-100, ura3, Yep13-pet127 (Δ48–215) | **4f** |
| AA40316 | MATa, ade2-1, leu2-3, trp1-1, his3-11,15, can1-100, ura3, Yep13-pet127 (1–215) | **4f** |
| AA39617 | MATa, ade2-1, leu2-3, trp1-1, his3-11,15, can1-100, ura3, Yep13-pet127-nd | **4f** |
| AA40317 | MATa, ade2-1, leu2-3, ura3, trp1-1, his3-11,15, can1-100, YEP13, pRS426 | **5a, 5b, S8a, S8b** |
| AA40319 | MATa, ade2-1, leu2-3, ura3, trp1-1, his3-11,15, can1-100, YEP13-PET127, pRS426 | **5a, 5b, S8a, S8b** |
| AA40318 | MATa, ade2-1, leu2-3, ura3, trp1-1, his3-11,15, can1-100, YEP13, pRS426-MTS-3xV5-RPO41 | **5a, 5b, S8a, S8b** |
| AA40320 | MATa, ade2-1, leu2-3, ura3, trp1-1, his3-11,15, can1-100, YEP13-PET127, pRS426-MTS-3xV5-RPO41 | **5a, 5b, S8a, S8b** |
| AA36029 | MATa, ade2-1, leu2-3, ura3, trp1-1, HIS3+, can1-100 | **5c, S11** |
| AA40931 | MATa, ade2-1, leu2-3, ura3, trp1-1, can1-100, rpo41::KANmx6, Rho0 | **5c, S11** |
| AA40974 | MATa, ade2-1, trp1-1, leu2-3, can1-100, ura3::RPO41preseq-3xV5-RPO41::URA3, HIS3+ | **5c, S11** |
| AA40937 | MATa, ade2-1, trp1-1, his3-11,15, can1-100, ura3::RPO41preseq-3xV5-RPO41::URA3, rpo41::KANmx6, HIS+ | **5c, S11** |

(*Continued*)

**Table 1.** (Continued)

| Strain Number | Relevant Genotype | Fig |
|---|---|---|
| AA40972 | *MATa, ade2-1, trp1-1, leu2-3, can1-100, ura3::RPO41preseq-3xV5-RPO41-EI::URA3, HIS3+* | **5c, S11** |
| AA40966 | *MATa, ade2-1, trp1-1, leu2-3, can1-100, ura3::RPO41preseq-3xV5-RPO41-EI::URA3, rpo41::KANmx6, HIS3+* | **5c, S11** |
| AA41033 | *MATa, ade2-1, trp1-1, his3-11,15, can1-100, ura3::RPO41preseq-3xV5-RPO41::URA3, rpo41::KANmx6, LEU2+, Rho- (ORI5-1)* | **5d** |
| AA41017 | *MATa, ade2-1, trp1-1, his3-11,15, can1-100, ura3::RPO41preseq-3xV5-RPO41-EI::URA3, rpo41::KANmx6, LEU2+, Rho- (ORI5-1)* | **5d** |
| AA40982 | *MATa, ade2-1, trp1-1, his3-11,15, can1-100, ura3::RPO41preseq-3xV5-RPO41::URA3, rpo41::KANmx6, LEU2+, Rho0* | **5d S1c** |
| AA41006 | *MATa, ade2-1, trp1-1, his3-11,15, can1-100, ura3::RPO41preseq-3xV5-RPO41-EI::URA3, rpo41::KANmx6, LEU2+, Rho0* | **5d** |
| AA40938 | *MATalpha, ade2-1, trp1-1, his3-11,15, can1-100, ura3::RPO41preseq-3xV5-RPO41::URA3, rpo41::KANmx6, HIS+* | **5d. S1c** |
| AA40967 | *MATalpha, ade2-1, trp1-1, leu2-3, can1-100, ura3::RPO41preseq-3xV5-RPO41-EI::URA3, rpo41::KANmx6, HIS3+* | **5d** |
| AA41640 | *MATa, ade2-1, leu2-3, trp1-1, his3-11,15, can1-100, Pet127-6xHA::HIS3, TOM70-GFP::KanMX* | **S7** |
| AA41642 | *MATa, ade2-1, leu2-3, trp1-1, his3-11,15, can1-100, Pet127(Δ48–215)-6xHA::HIS3, TOM70-GFP::KanMX* | **S7** |
| AA41643 | *MATa, ade2-1, leu2-3, trp1-1, his3-11,15, can1-100, Pet127(1–215)-6xHA::HIS3, TOM70-GFP::KanMX* | **S7** |
| AA41809 | *MATa, ade2-1, leu2-3, trp1-1, his3-11,15, can1-100, ura3, pYX223-mtGFP* | **S9a, S9b** |
| AA39631 | *MATa, ade2-1, leu2-3, trp1-1, his3-11,15, can1-100, ura3, pRS423* | **S9a, S9b, S9c** |
| AA39838 | *MATa, ade2-1, leu2-3, trp1-1, his3-11,15, can1-100, ura3, pRS423-pGal-PET127-3xV5* | **S9a, S9b, S9c** |
| AA40159 | *MATa, ade2-1, leu2-3, trp1-1, his3-11,15, can1-100, ura3, pRS423-pGal-pet127-nd-3xV5* | **S9a, S9b, S9c** |
| AA40237 | *MATa, ade2-1, leu2-3, trp1-1, his3-11,15, can1-100, ura3, pRS423-pGal-pet127Δ(48–215)-3xV5* | **S9a, S9b, S9c** |
| AA40335 | *MATa, ade2-1, leu2-3, trp1-1, his3-11,15, can1-100, ura3, pRS423-pGal-pet127(1–215)-3xV5* | **S9a, S9b, S9c** |
| AA41810 | *MATalpha, ade2-1, leu2-3, ura3, HIS3+, can1-100, trp1-1 LEU2:: pGal1-PET127 (Single copy integration) HS ORI5-1 rho-* | **S9d** |
| AA41811 | *MATalpha, ade2-1, leu2-3, ura3, HIS3+, can1-100, trp1-1 LEU2:: pGal1-PET127 (Single copy integration) HS ORI5-2 rho-* | **S9d** |

## EtBr Treatment (Generation of *rho0* and *HS rho-*)

For generation of *rho0* cells of a particular strain, cells were grown in YEPD + 10µg/ml ethidium bromide for 3 days, diluted back each morning. After treatment cells were streaked to single colonies and lack of mtDNA was confirmed via DAPI staining at 2µg/ml. Generation of *HS* alleles was performed by growing log phase *rho+* cells in 5µg/ml EtBr for 100 minutes and plated on YEPD plates to ~200 cells per plate.

## Screen

Rho+ cells were transformed with Yep13 high-copy library [45] and were plated on plasmid selection media. Colony plates were replica plated to lawns of *HS ORI5-1*, and further replica plated to diploid selection medium and replica plated again to YEPD with no additional

adenine. Identified colonies were selected from the initial transformation plate and the assay was repeated on those colonies. Plasmids from colonies which scored both times were Sanger sequenced using Yep13 sequencing primers. The resulting sequences were identified via NCBI blast [63].

## Cytoduction

HS strains were transformed with a matA cassette and a Pringle deletion cassette for matAlpha and mated with *rho0* karyogomy defective mtDNA shuttling strain AA39257 and plated on YEPD + 10μg/ml Cycloheximde to select for homozygous *cyh2Δ* present only in haploid AA39257. The resulting colonies were screened for cells containing mtDNA via DAPI indicating transfer of HS mtDNA from the HS strain to the AA39257 parent. The resulting cells were mated with *rho0* recipient strains, and plated on SD-Arg+ 50mg/ml Canavanine to select for homozygous *can1-1* allele present only in the haploid recipient strains. The resulting colonies were checked for mtDNA by staining with 2μg/ml DAPI and confirmed to be haploid by mating complementation on minimal media.

## Mating assay and temperature shifts

Log phase growing cells were mated by mixing 2.2 x $10^7$ cells from each parent into 1ml YEPD at room temperature for 5 hours. Cells were diluted 1:1000 and plated on diploid selection plates (SD-His-Leu or SD-His-Ura-Leu) for 48hrs at either 30°C, 34°C, or 37°C and then replica plated to YEPG (1% yeast extract, 2% peptone, 3% glycerol) plates and placed at the same temperature for 72 hours. Percent *rho+* cells was determined by 100*#colonies growing on YEPG/#colonies growing on diploid selection. A colony was considered growing if any subregion of the colony was growing.

## Rpo41-V5-EI temperature sensitivity

Mid-log (~1.1 x $10^7$ cells/ml) cultures were diluted to 1.1 x $10^7$ cells/ml and five-fold serially diluted 5 times and 4 μl was plated on either YEPD, YEPG, or SD-His and placed at either 30°C, 34°C, or 37°C for 2 days.

## Cloning and strain construction

Strains were constructed using standard gene integration or tagging strategies [64]. Plasmids were constructed using standard cloning techniques and described in Table 2 [65].

## DNA isolation

DNA for qPCR and colony PCRs was isolated using a modified smash and grab protocol from [66] with RNAse treatment. The mtDNA isolation for SMRT sequencing was done similarly using two phenol chloroform extractions on pellets enriched for mitochondria using the mitochondrial isolation technique below.

## SMRT sequencing

mtDNA isolated as above was sequenced using SMRT sequencing technology on Pacific Biosciences Sequel II [67]. 10kb insert size was selected for on 0.75% agarose Blue Pippin cassettes obtaining 12-13kb mean length inserts and mean read length of 6-7kb. Long reads were computationally divided into 50nt fragments and mapped using BWA-mem onto SGD sacCer3 genome wildtype reference and viewed on IGV [68].

**Table 2. List of Plasmids Used in this Study.**

| Plasmid Number | Description | Fig |
|---|---|---|
| pAA44 | YEp13 | 2, 3b, 3c, 4f, 5a, 5b, S2, S3b, S3c, S5, S8a, S8b |
| pAA2582 | YEp13-Erp4 | 3b |
| pAA2581 | YEp13-Pet127 | 3b, 3c, 4f, 5a, 5b, S3b, S3c, S5, S8a, S8b |
| 20.4 | YEp13-[3'RTS1, YOR015W, ERP4, PET127, 5'ROD1] | 3b, S3a |
| 30.2 | YEp13-[3'RTS1, YOR015W, ERP4, 5'PET127] | S3a |
| 39.1 | YEp13-[3'CHS1, ARS1414, DUG3, YNL190W, SRP1] | S3a |
| 39.4 | YEp13-[3'SEC27, MRM2, RPL1b, 5'PCL10] | S3a |
| pAA2758 | pGEX-6P-1-V5-Rpo41 | 4b |
| pAA2778 | pET15b-Pet127-6xHis (Derived from pET15b Novagen (EMD Millipore)) | 4b |
| pAA2058 | pGEX-6P-1 (Amersham) | 4b |
| pAA2794 | pET15b-Pet127-N504-6xHis | 4b |
| pAA2806 | pET15b-Pet127-C1764-6xHis | 4b |
| pAA2703 | pRS306-Rpo41-internal-3xV5 | 4a, 4d, 5c, 5d, S10a, S10b, S11 |
| pAA2782 | pNH605-Pet127-nd-3xV5 | 4d, S7 |
| pAA2832 | pNH605-Pet127$^{\Delta48\text{-}215}$-3xV5 | 4d, S7 |
| pAA2842 | pNH605-Pet127$^{1\text{-}215}$-6xHA-His3 | 4d, S7 |
| pAA2713 | YEp13-Pet127-nd | 4f |
| pAA2833 | YEp13-Pet127$^{\Delta48\text{-}215}$ | 4f |
| pAA2835 | YEp13-Pet127$^{1\text{-}215}$ | 4f |
| pAA2837 | pRS426-RPO41-internal-3xV5 | 5a, 5b, S8a, S8b |
| pAA2924 | pRS306-Rpo41-EI-internal-3xV5 | 5c, 5d, S11 |

## Colony PCR

Strains were mated as in the quantitative mating assay. Whole individual diploid colonies were scraped off plates and DNA was isolated from them using the DNA isolation protocol. DNA was normalized across samples and added to PCR reactions using the colony PCR primer sets in Table 3. PCRs were run on either 1% or 2% (ATP9) agarose gel made with 1x TAE (40mM Tris base, 20mM Acetic Acid, 1mM EDTA) and 0.1μg/ml EtBr for 25 minutes at 130V constant voltage in 1 x TAE.

**Table 3. List of Primers Used in this Study.**

| Primer Set | Forward Primer | Reverse Primer |
|---|---|---|
| Yep13 Backbone | TTCGCTACTTGGAGCCACTAT | ATCGGTGATGTCGGCGATATA |
| *ACT1* qPCR set | GTACCACCATGTTCCCAGGTATT | CAAGATAGAACCACCAATCCAGA |
| *COX3* qPCR set | TCCATTCAGCTATGAGTCCTGA | CTGCGATTAAGGCATGATGA |
| *COX2* qPCR set | GTGGTGAAACTGTTGAATTTGAATC | AGCAGCTGTTACAACGAATCTA |
| *ATP9* qPCR set | ATTGGAGCAGGTATCTCAACAA | GCTTCTGATAAGGCGAAACC |
| *ORI* qPCR set | ATAGGGGGAGGGGGTGGGTGAT | GGGACCCGGATATCTTCTTGTTTATC |
| Nuclear Colony PCR Set (*PET127* locus) | GCGCGTTTCCGTCAATGCC | TTTCAGTAGATTAATCGCCTTGTCC |
| *ORI* Colony PCR set | ATAGGGGGAGGGGGTGGGTGAT | GGGACCCGGATATCTTCTTGTTTATC |
| *COX2* Colony PCR set | CAGCAACACCAAATCAAGAAGG | ATGACCTGTCCCACACAAC |
| *COX3* Colony PCR set | GACACATTTAGAAAGAAGTAGACATCAAC | GACTCCTCATCAGTAGAAGACTACG |
| *ATP9* Colony PCR set | ATTGGAGCAGGTATCTCAACAA | GCTTCTGATAAGGCGAAACC |
| *COX1 Nterm* Colony PCR set | TAGCTGCACCTGGTTCACAA | CCTCTTTCAGTTGATCCCTCAC |
| *COX1 Cterm* Colony PCR set | ACTTTCTTCCCCTCCGAATC | CCTGCGGATTGTCCATACTT |

## RNA Isolation and qPCR

RNA was isolated from 4.4 x $10^7$ cells using acid phenol and purified using an RNeasy kit (Qiagen) with on column DNAse treatment (Qiagen). Reverse transcription was performed on 750ng of RNA using SuperScript III First-Strand Synthesis SuperMix (Life Technologies) and qPCR was performed using SYBR Premix Ex Taq (Tli RNaseH Plus) or the equivalent TB Green Premix Ex Taq (Tli RNaseH Plus) from TaKaRa. Signals were normalized to *ACT1* levels and normalized to the control average. Primers are described in Table 3.

## RNA sequencing

Isolated total yeast RNA as above were processed using Ribozero rRNA removal for Yeast (Illumina). For *pet127-nd* RNA profiling, sequencing was performed on NextSeq500 with 75 + 75 bases pair-end run with 6 + 6 nucleotide indexes. Pair end sequencing reads were mapped to sacCer3 reference genome using star/2.5.3a [69]. The regions of interest were defined using the bed file format. The coverage sub-command of bedtools/2.26.0 was applied to calculate the number of sequences mapped to the specific regions of the reference genome using alignment bam files. The counts of all samples were merged to a matrix with each sample per column and each location per row using MIT IGB in house tool. The regions were defined by the boundaries in Table 4 modified from [7].

For the *pGal-PET127* RNA sequencing experiment, sequencing was performed on HiSeq2000 with 40 bases single end run with 8 + 8 nucleotide indexes [70]. Single end sequencing reads were mapped to sacCer3 reference genome using star/2.5.3a. rsem/1.3.0 was applied for gene level counting, fpkm and tpm calculation. The raw counts, fpkm, and tpm values of each gene in all sample were merged into three corresponding matrices using IGB in house tools. The matrices were formatted as each sample per column and each gene per row. Hierarchical clustering was performed using TIBCO Spotfire 7.11.1 based on log2(fpkm+1) of the expressed coding genes. Differential expression comparisons between 5 hour and 0 hour under different conditions were carried out using Deseq2 1.10.1 under r/3.2.3. with raw counts as input.

## Mitochondria isolation

Mitochondrial enrichment protocol modified from [71]. Cells were grown into logarithmic phase growth, transferred into DTT buffer (0.1 M Tris pH 9.4, 10 mM DTT) to shake for 20 minutes at 30°C, and transferred into zymolyase buffer (1.2M sorbitol, 20mM $K_2HPO_4$ pH 7.4) with the addition of 1% zymolyase 100T or equivalent for 1 hour at 30°C to digest the cell wall. Cells were lysed by dounce homogenization 20 strokes in homogenization buffer (0.6M sorbitol, 10mM Tris pH 7.4, 1mM EDTA, 0.2% BSA no fatty acid, 1mM PMSF). Mitochondria were isolated by differential centrifugation 5 minutes at 1200g, the resulting supernatant was spun for 5 minutes at 2000g, and the resulting supernatant was spun at 17500g for 15 minutes all at 4°C. The resulting pellet was resuspended in SEM buffer (0.25M sucrose, 10mM MOPS KOH pH 7.2, and 1mM EDTA). For assessing whether proteins are mitochondrial, 3μg of enriched mitochondria was treated with 50μg/ml proteinase K for 5mins at 37°C. The reaction was stopped by addition of TCA to 12.5%.

## Coimmunoprecipitation assay

Cells were grown in YEPD to 1.1 x $10^7$ cells /ml. 1.1 x $10^9$ cells were collected and frozen. Cells were lysed with a FastPrep-24 Classic (MP Biomedicals, Speed 6.5, 60s, 10 cycles) with 200μl NP40 buffer [50mM Tris pH7.5, 150mM NaCl, 1% Np40 (IGEPAL) and Halt Protease Inhibitor Cocktail (Thermo Fisher Scientific)]. Lysates were brought up to 1.5ml with NP40 buffer

**Table 4. Bin Boundaries for *pet127-nd* RNA Sequencing.**

| Description | Start | Start |
|---|---|---|
| RPM1 to UGG Proline tRNA | 1 | 730 |
| UGG Proline tRNA | 731 | 802 |
| UGG Proline tRNA to ORI1 | 803 | 4011 |
| ORI1 | 4012 | 4312 |
| ORI1 to 15S rRNA | 4313 | 6545 |
| 15S rRNA | 6546 | 8194 |
| 15S rRNA to UCA Tryptophan tRNA | 8195 | 9373 |
| UCA tryptophan tRNA | 9374 | 9444 |
| UCA tryptophan tRNA to ORI8 | 9445 | 12509 |
| ORI8 | 12510 | 12780 |
| ORI8 to COX1 | 12781 | 13817 |
| COX1 | 13818 | 26701 |
| COX1 to ATP8 | 26702 | 27665 |
| ATP8 | 27666 | 27812 |
| ATP8 to ATP6 | 27813 | 28486 |
| ATP6 | 28487 | 29266 |
| ATP6 to ORI7 | 29267 | 30219 |
| ORI7 | 30220 | 30594 |
| ORI7 to ORI2 | 30595 | 32230 |
| ORI2 | 32231 | 32501 |
| ORI2 to UUC glutamate tRNA | 32502 | 35372 |
| UUC glutamate tRNA | 35373 | 35444 |
| UUC glutamate tRNA to COB | 35445 | 36539 |
| COB | 36540 | 43647 |
| COB to ORI6 | 43648 | 44888 |
| ORI6 | 44889 | 45225 |
| ORI6 to ATP9 | 45226 | 46722 |
| ATP9 | 46723 | 46953 |
| ATP9 to UGA serine tRNA | 46954 | 48200 |
| UGA serine tRNA | 48201 | 48290 |
| UGA serine tRNA to VAR1 | 48291 | 48900 |
| VAR1 | 48901 | 50097 |
| VAR1 to ORI3 | 50098 | 54566 |
| ORI3 | 54567 | 54840 |
| ORI3 to ORI4 | 54841 | 56566 |
| ORI4 | 56567 | 56832 |
| ORI4 to 21S rRNA | 56833 | 58008 |
| 21S rRNA | 58009 | 62447 |
| 21S rRNA to UGU threonine tRNA | 62448 | 63861 |
| UGU threonine tRNA | 63862 | 63934 |
| UGU threonine to GCA cysteine tRNA | 63935 | 64414 |
| GCA cysteine tRNA | 64415 | 64487 |
| GCA cysteine tRNA to GUG histidine tRNA | 64488 | 64596 |
| GUG histidine tRNA | 64597 | 64667 |
| GUG histidine tRNA to UAA leucine tRNA | 64668 | 66094 |
| UAA leucine tRNA | 66095 | 66176 |
| UAA leucine tRNA to UUG glutamine tRNA | 66177 | 66209 |

(*Continued*)

**Table 4.** (Continued)

| Description | Start | Start |
|---|---|---|
| UUG glutamine tRNA | 66210 | 66282 |
| UUG glutamine tRNA to UUU lysine tRNA | 66283 | 67060 |
| UUU lysine tRNA | 67061 | 67132 |
| UUU lysine tRNA to UCU arginine tRNA | 67133 | 67308 |
| UCU arginine tRNA | 67309 | 67381 |
| UCU arginine tRNA to UCC glycine tRNA | 67382 | 67467 |
| UCC glycine tRNA | 67468 | 67539 |
| UCC glycine tRNA to GUC aspartate tRNA | 67540 | 68321 |
| GUC aspartate tRNA | 68322 | 68393 |
| GUC aspartate tRNA to GCU Serine tRNA | 68394 | 69202 |
| GCU Serine tRNA | 69203 | 69285 |
| GCU Serine tRNA to ACG arginine tRNA | 69286 | 69288 |
| ACG arginine tRNA | 69289 | 69359 |
| ACG arginine tRNA to UGC alanine tRNA | 69360 | 69845 |
| UGC alanine tRNA | 69846 | 69918 |
| UGC alanine tRNA to GAU Isoleucine tRNA | 69919 | 70161 |
| GAU Isoleucine tRNA | 70162 | 70234 |
| GAU Isoleucine tRNA to GUA tyrosine tRNA | 70235 | 70823 |
| GUA tyrosine tRNA | 70824 | 70908 |
| GUA tyrosine tRNA to GUU Asparagine tRNA | 70909 | 71432 |
| GUU Asparagine tRNA | 71433 | 71504 |
| GUU Asparagine tRNA to CAU methionine tRNA 1 | 71505 | 72631 |
| CAU methionine tRNA 1 | 72632 | 72705 |
| CAU methionine tRNA 1 to COX2 | 72706 | 73757 |
| COX2 | 73758 | 74513 |
| COX2 to GAA phenylalanine tRNA | 74514 | 77430 |
| GAA phenylalanine tRNA | 77431 | 77502 |
| GAA phenylalanine tRNA to UAG threonine tRNA | 77503 | 78088 |
| UAG threonine tRNA | 78089 | 78162 |
| UAG threonine tRNA to UAC valine tRNA | 78163 | 78532 |
| UAC valine tRNA | 78533 | 78605 |
| UAC valine tRNA to COX3 | 78606 | 79212 |
| COX3 | 79213 | 80022 |
| COX3 to ORI5 | 80023 | 82328 |
| ORI5 | 82329 | 82600 |
| ORI5 to CAU methionine tRNA 2 | 82601 | 85034 |
| CAU methionine tRNA 2 | 85035 | 85107 |
| CAU methionine tRNA 2 to RPM1 | 85108 | 85294 |
| RPM1 | 85295 | 85777 |
| RPM1 to UGG Proline tRNA | 85778 | 85779 |

and were pelleted by centrifugation at 20,000g for 10 minutes at 4C. 1% of sample was taken for Input and boiled with 10μl of 3x SDS sample buffer, and the rest of the lysates were brought to 0.2% BSA. Clarified sample was added to 20μl of Anti-V5 agarose affinity gel antibody beads (Sigma) and incubated for 2 hours at 4C. Beads were washed 3 times with Np40 buffer with Protease inhibitor and 2 times in NP40 buffer without inhibitors. To elute, 30μl of 1.5x SDS sample buffer was added to the beads and samples were boiled for 5 minutes.

## Immunoblotting

Samples in SDS sample buffer were run on 8% polyacrylamide gels in SDS running buffer (0.1918M glycine, 0.0248M Tris base, 0.1%SDS), or gradient gels in MES running buffer (2.5mM MES, 2.5 mM Tris Base, 0.005% SDS, 0.05mM EDTA pH7.3), transferred to PVDF membranes via semi-dry transfer, washed with PBST (10mM, 2.7mM potassium chloride, 137mM sodium chloride, and 1.76mM potassium phosphate. pH 7.4 + 0.1% Tween-20), blocked with PBST and 3% milk for 1 hour and probed overnight in 1xPBS with 1% milk, 1% BSA, 0.1% Sodium Azide and with one of the following primary antibody dilutions: Mouse Anti-HA (1:2000; HA.11, Biolegend), Mouse Anti-V5 (1:2000; R960-25, Thermo-Fisher Scientific), Mouse Anti-VDAC/Porin (1:1000; 16G9E6BC4/ab110326, Abcam), Mouse Anti-6xHis (1:5000, ab18184, Abcam), Mouse Anti-GST (1:5000, ab19256, Abcam), Mouse Anti-PGK (1:5000; 22C5D8, Thermo-Fisher Scientific), Mouse Anti-COX4 (1:1000; ab110272, Abcam), Mouse Anti-GFP (1:1000; JL-8, Clontech). Ponceau S (Abcam) was applied before blocking for 15 minutes and destained for 30 minutes with distilled water. Blots were washed three times with PBST and probed with secondary antibodies diluted in PBS with 1% milk and 1% BSA for one hour at 4˚C. Secondary antibody dilutions were Anti-mouse HRP 1:10,000 and Anti-mouse HRP TrueBlot 1:1000 (for coimmunoprecipitation experiments). Membranes were washed three times in PBST and blots were imaged using the ECL Plus system (GE healthcare).

## Bacterial expression and binding and truncations

BL21 Bacteria containing expression plasmids, described in Table 5, were grown in LB+AMP (1% Difco Bacto Tryptone, 0.5% Difco Yeast Extract, 1% NaCl, 10mM Tris pH 7.5, Amp 0.1 mg/ml) overnight at 37˚C. Cells were diluted to $2.66 \times 10^8$ cells/ml in LB + AMP and grown at 37˚C for 1h, IPTG was added to 0.5 mM final concentration and cells were allowed to express for 5h at 37˚C. Cells were spun at 22K g for 20 minutes at 4˚C and resuspended to $1.33 \times 10^{10}$ cells/ml in PBS + PI [cOmplete, mini, EDTA-free Protease Inhibitor Cocktail (Millipore-Sigma) 1/50mls] + 0.3% BSA. Cells were lysed via French press. GST expressing samples were added to Glutathione beads (pre-washed in PBS+PI). Beads were incubated for 2h at 4˚C and spun down at 500 rcf for 2 minutes at 4˚C. Beads were washed 3 times with cold NP40 buffer + PI + BSA 0.3%. Equal amounts of non-GST expressing samples were added to the beads. Beads were incubated for 2h at 4˚C and spun down at 500 rcf for 2 minutes at 4˚C. Beads were washed 3 times with cold NP40 buffer + PI +BSA 0.3% and washed 2 times with NP40 buffer. Protein was eluted by adding 2x SDS Sample buffer equal in volume to beads and boiling for 5 minutes.

## Diauxic shift experiments

For the glucose to ethanol shift, cells were grown overnight in YEPD medium and diluted back to $3.3 \times 10^7$ cells/ml in YEPD. Cells were grown for 12 hours with $2.2 \times 10^7$ cells collected

**Table 5. List of Bacterial Strains Used in this Study.**

| Strain number | Relevant Genotype | Fig |
|---|---|---|
| AAE568 | BL21 pGEX-6P-1-V5-Rpo41 (pAA2758) | 4b |
| AAE569 | BL21 pET15b-PET127-6xHIS (pAA2778) | 4b |
| AAE570 | BL21 pGEX-6P-1 (pAA2058) | 4b |
| AAE571 | BL21 pET15b-PET127N504-6xHis (pAA2794) | 4b |
| AAE572 | BL21 pET15b-Pet127-C1764-6xHis (pAA2806) | 4b |

every 2 hours. For the ethanol to glucose shift, A 2.2 x $10^7$ cells from an overnight culture were collected and then the culture was diluted back to 3.3 x $10^6$ cells/ml in YEPD. 2.2 x $10^7$ cells were collected 3h, 5h, and 7h later.

## Medium swap experiment

Cells were grown overnight in YEPD medium and diluted back to 1.1x$10^7$ cells/ml and grown until 1.1x$10^7$ cells/ml. 2.2 x $10^7$ cells were collected and then equal amounts of cells were collected using filters (Supor membrane disc filters, Pall laboratory, 47mm, 0.8μM, VWR) and released into YEPD, YEPG (2% Glycerol), YEPA (1% yeast extract, 2% peptone, 2% Potassium Acetate). YEPE (1% yeast extract, 2% peptone, 2% ethanol) at 1.1x$10^7$ cells/ml. Cells were grown for 4 hours and 2.2 x $10^7$ cells were collected and immunoblotted.

## Supporting information

**S1 Fig. Generation of *rho-* and *rho0* Colonies over Time upon Addition of EtBr.** A. 5 μg/ml EtBr was added to *rho+* cells and cells were collected every 30 minutes for 180 minutes. Colonies were assessed for respiration. Percent *rho-* or *rho0* colonies was calculated by the formula 100 x [1- (respiratory colonies/total colonies)]. B. Diagram of screen for HS alleles. *Rho+* cells were treated with 5μg/ml EtBr for 100 minutes to create a mixed mtDNA population, plated to single colonies on YEPD (1% yeast extract, 2% peptone, 2% glucose), mated with lawns of *rho + yeast* on YEPD, selected for diploids on SD-His-Leu, and plated on YEPD plates with no added adenine. Red colonies on the low adenine YEPD plate are *rho+* and white colonies are *rho-* or *rho0*. White colonies were taken for further analysis. C. HS candidates tested for mtDNA inheritance bias by quantitative mating assay. HS candidates or a *rho0* control were mated with a *rho+* tester and selected for diploids colonies which were assessed for *rho+* mtDNA.
(TIF)

**S2 Fig. PCR detection limits for *Rho+* loci.** Genomic DNA was isolated from *rho+* or *HS ORI5-1* parent patches and normalized. *Rho+* genomic DNA was diluted into *HS ORI5-1* genomic DNA in five-fold serial dilutions. PCRs of using primer sets recognizing mitochondrial loci were performed from the serial dilutions and run on agarose gels containing ethidium bromide. The first lane contains the ladder and the last lane is a no DNA control.
(TIF)

**S3 Fig. *PET127* is the Cause of Suppression by High-Copy Plasmids.** A. Inserts of four plasmids obtained from the high-copy suppressor screen were PCRed using Q5 (New England Biolabs) and Sanger sequenced using the Yep13 backbone primer set. The resulting sequences were identified via NCBI blast [63] and the genomic boundaries of each plasmid insert are as indicated. Parentheses indicate annotated genes contained within the genomic region with 3' or 5' indicating that the genic region was cut off and only the 3' or 5' end of the gene was present. B. The suppressive capability of the high-copy *PET127* plasmid was tested on *HS ORI5* alleles cytoduced into a common recipient strain so as to confirm that the extent of high-copy *PET127* suppression is determined by the HS allele and not by a possible nuclear mutation. C. Typical plates following quantitative inheritance assay. "Total Diploid" images taken 2 days after plating on SD-His-Leu diploid selection medium. "Respiratory" images taken 3 days after replica plating to YEPG. Inset showing colony section growth.
(TIF)

**S4 Fig. Mitochondrial RNA Sequencing Profile of *pet127-ND* Allele Mimics that of *pet127Δ*.** Mitochondrial gene expression analysis of A. Intragenic mtDNA regions with and

without *COX1* as *COX1* has much higher expression relative to the other mitochondrial genes. B. Non-coding RNA regions: *tRNA* and *ORI*. C. Intergenic mtDNA regions. * indicates a discovery in both comparisons *WT* to *pet127-ND/pet127Δ* and *WT* to *pet127Δ* via multiple unpaired T-tests with Two-stage step-up (N = 3). "x" indicates discovery in only *WT* to *pet127Δ*. + indicates discovery in only *WT* to *pet127-ND/pet127Δ*. No discoveries were identified in *WT* to *WT/pet127-ND*. Region delineations can be found in Table 4.
(TIF)

**S5 Fig. The *pet127-nd* Suppresses HS Biased Inheritance.** Quantitative mtDNA inheritance assay with high-copy *pet127-nd*. Significance was determined using one-way ANOVA separately on each *HS* and *rho0* cross with means compared to the vector control using Dunnett's multiple comparison's test (N = 3). **** indicates adjusted P-value less than 0.0001. All other comparisons were not significantly different.
(TIF)

**S6 Fig. Diagram of tested *PET127* truncation alleles.** Diagram of tested *PET127* truncation constructs in the bacterial expression and binding assay. First row is full length *PET127* with predicted mitochondrial targeting sequence. Second row is with the mitochondrial targeting sequence replaced with 6xHis. All the following rows are truncation alleles to scale. Columns indicate whether the allele was able to express in bacteria or subsequently bind with Rpo41 as in Fig 4B.
(TIF)

**S7 Fig. *PET127* truncation alleles are able to get into the mitochondrion.** Cycling mutant *PET127* and *TOM70-GFP* carrying cells in YEPD were lysed by zymolyase treatment and Dounce homogenization. Mitochondria was enriched by differential centrifugation and 3μg of mitochondria was treated with 50μg/ml proteinase K for 5mins at 37˚C. The reaction was stopped by addition of TCA to 12.5%. Samples were immunoblotted as indicated. A low exposure was used for Pet127$^{1\text{-}215}$-HA because it was more abundant than Pet127-HA and Pet127$^{\Delta48\text{-}215}$-HA. Arrows indicate the expected size of Pet127 full length and Pet127$^{\Delta48\text{-}215}$-HA respectively.
(TIF)

**S8 Fig. *PET127* Overexpression Inhibits *RPO41* Transcription.** A. RT-qPCR of either *COX3*, *COX2* or *ATP9* divided by *ACT1* RNA and normalized by the qPCR of the respective gene divided by *ACT1* DNA. RT-qPCRs were performed on the high-copy *PET127* high-copy *RPO41* combination strains collected at high cell density to increase respiration. The left RT-qPCRs are normalized to the *Vectors* averages. The right column is the same data visualized as a log2 fold change relative to the *Vectors* control. Significance was determined using one-way ANOVA comparing all values with each other using Tukey's multiple comparison's test (N = 2). **** indicates adjusted P-value less than 0.0001, *** indicates an adjusted P-value less than 0.001, ** indicates an adjusted P-value less than 0.01, * indicates an adjusted P-value less than 0.05. B. qPCR of either *COX3*, *COX2*, or *ATP9* divided by *ACT1* qPCR on DNA isolated from high-copy *PET127* high-copy *RPO41* combination strains. No comparisons were determined as significantly different. Significance was determined using one-way ANOVA comparing all values with each other using Tukey's multiple comparison's test (N = 2).
(TIF)

**S9 Fig. *pGal-PET127* Alleles Require Rpo41 Binding Region to Alter Mitochondrial RNA but not *ORI* Transcription.** A. *COX3* RNA RT-qPCR, *COX3* DNA qPCR, or *COX3* RNA RT-qPCR divided by *COX3* DNA qPCR on high-copy *pGal-Pet127* alleles in *rho+* cells normalized

to *ACT1* RNA and/or DNA 0, 3, and 5 hours after addition of galactose. Significance was determined using one-way ANOVA separately for each time point with means compared to either the vector control or the *pGal-mtGFP* control using Dunnett's multiple comparison's test. *** indicates an adjusted P-value between 0.001 and 0.0001. * indicates an adjusted P-value between 0.05 and 0.01. All other comparisons were not significantly different. B. Gene expression analysis via RNA sequencing of *PET127* or mitochondrial genic transcripts in cells carrying high-copy *pGal-Pet127* alleles. Values indicate average log2 fold change between 0 hours and 5 hours after addition of galactose for three independent experiments. C. *ORI* RNA RT-qPCR, *ORI* DNA qPCR, or *ORI* RNA RT-qPCR divided by *ORI* DNA qPCR on high-copy *pGal-Pet127* alleles in *rho+* cells normalized to *ACT1* RNA and/or DNA 0, 3, and 5 hours after addition of galactose. Significance was determined using one-way ANOVA separately for each time point with means compared to either the vector control or the *pGal-mtGFP* control using Dunnett's multiple comparison's test (N = 3). **** indicates an adjusted P-value less than 0.0001. ** indicates an adjusted P-value between 0.01 and 0.001. All other comparisons were not significantly different. D. *ORI* RNA RT-qPCR on *HS ORI5-1* and *HS ORI5-2* with *pGal-Pet127* RNA normalized to *ACT1* 0, 3, and 5 hours after addition of galactose. Significance was determined using student's T-test comparing equivalent time points between strains containing either the same *ORI* or *pGal-PET127 allele* (N = 2 or 3). * indicates a P-value between 0.05 and 0.01. All other comparisons were not significant.
(TIF)

**S10 Fig. Pet127 Protein is Less Abundant in Respiratory Conditions and More Abundant in Glucose Medium.** A. Left panel: *PET127-HA V5-RPO41* cells were grown to high cellular density in YEPD medium and samples were collected and immunoblotted for HA, V5 and VDAC. Right panel: A high density culture of *PET127-HA V5-RPO41* cells in YEPD medium was diluted at 0h to 3.3 x 10$^6$ cells /ml in YEPD medium and samples were collected over time and immunoblotted for HA, V5 and VDAC. B. *PET127-HA V5-RPO41* cells were grown in YEPD medium to mid log, a sample was taken and then the culture was split into YEP medium containing either 2% Glucose, 2% Glycerol, 2% Potassium Acetate, or 2% Ethanol. Samples were taken after 4 hours and samples were stained with Ponceau S and immunoblotted for HA, V5 and VDAC.
(TIF)

**S11 Fig. Synthetic lethality of *rpo41-EI* at 37˚C on synthetic dropout medium.** Five-fold serial dilutions of strains containing *RPO41* alleles beginning at 1.1 x 10$^7$ cells/ml plated on SD-His and incubated at either 30˚C, 34˚C, or 37˚C for 2 days.
(TIF)

## Acknowledgments

We thank Dmitriy Markov for insight about RPO41 allele design, the MIT Biomicro Center and Stuart Levine for the Illumina sequencing experiments and discussions about experimental design of the SMRT and mitochondrial RNA sequencing, Charlie Whittaker and Duanduan Ma of the Barbara K. Ostrom (1978) Bioinformatics and Computing Facility at the Koch Institute in the Swanson Biotechnology Center for sequencing analysis of the SMRT sequencing and RNA sequencing experiments respectively, Maria Zapp and Ellen Kittler of the UMMS Deep Sequencing and Molecular Biology Core Laboratories and the PacBio Core Enterprise for SMRT sequencing, Xiaoxue Zhou for taking the image of S11 Fig, Stephen P. Bell and Matthew Vander Heiden for mentorship and critical reading of the manuscript, and Hilla Weidberg, Xiaoxue Zhou, and John Replogle for critical reading of the manuscript.

## Author Contributions

**Conceptualization:** Daniel Corbi, Angelika Amon.

**Formal analysis:** Daniel Corbi.

**Funding acquisition:** Angelika Amon.

**Investigation:** Daniel Corbi.

**Methodology:** Daniel Corbi.

**Project administration:** Angelika Amon.

**Resources:** Angelika Amon.

**Supervision:** Angelika Amon.

**Writing – original draft:** Daniel Corbi.

**Writing – review & editing:** Daniel Corbi.

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
