## [Decision Letter · Decision Letter 0]

25 May 2021

Dear Dr Corbi,

Thank you very much for submitting your Research Article entitled 'Decreasing Mitochondrial RNA Polymerase Activity Reverses Biased Inheritance of Hypersuppressive mtDNA' to PLOS Genetics.

The manuscript was fully evaluated at the editorial level and by independent peer reviewers. The reviewers appreciated the attention to an important problem, but raised some substantial concerns about the current manuscript. Based on the reviews, we will not be able to accept this version of the manuscript, but we would be willing to review a much-revised version. We cannot, of course, promise publication at that time.

The reviewers found the identification of pet127 as a modulator of uniparental inheritance for select hypersuprressive petites interesting and were also intrigued by remnants of rho+ DNA in the resulting diploid progeny from the rho+ x HS rho- crosses.    They felt the manuscript could be significantly strengthened with more quantitative analysis describing the loss of rho+ DNA;  controls for mtDNA template abundance when assessing transcription;  and more precise wording so as not to overstate any of the conclusions.  

If you decide to revise the manuscript for further consideration at PLOS Genetics, please aim to resubmit within the next 60 days, unless it will take extra time to address the concerns of the reviewers, in which case we would appreciate an expected resubmission date by email to plosgenetics@plos.org.

[LINK]

We are sorry that we cannot be more positive about your manuscript at this stage. Please do not hesitate to contact us if you have any concerns or questions.

Yours sincerely,

David M. MacAlpine

Guest Editor

PLOS Genetics

Gregory P. Copenhaver

Editor-in-Chief

PLOS Genetics

Reviewer's Responses to Questions

**Comments to the Authors:**

Reviewer #1: When mitochondrial genomes undergo a significant deletion event but retain at least one active replication origin, crosses of these haploid cells with ρ+ haploid cells of the opposite mating type will rise <5% ρ+ diploid progeny, a phenotype termed “hypersuppressive”. Such research is fundamental because deleted mtDNA has a replication advantage over wild-type mtDNA in heteroplasmy within a cell.

People usually think that the hypersuppressive mtDNA has a replicative advantage over the rho+ mtDNA. Thus, “Hypersuppressive mtDNA exerts a toxic effect on rho+ mtDNA” is a new concept. For me, it is exciting.

However, after reading this article, I have to say the manuscript still needs improvement by adding more data before consideration for publication.

Major questions:

(1) How can HS mtDNA actively eliminate regions of the rho+ genome?

“regions of rho+ mtDNA persisted after hypersuppressive takeover, indicating that hypersuppressive preferential inheritance may partially be due to the active destruction of rho+ mtDNA.”

We need evidence, not based on speculation. Recombination-deficient mutants reduce HS-biased inheritance by recombinase-driven mtDNA replication. We need more explanation on an imagined recombination DNA destruction mechanism.

(2) Is Pet127 an inducible protein? When does it inhibit the mitochondrial RNA polymerase RPO41.

Minor questions:

Page 5, line74~line 77:

In most cases, mating between yeast parents containing rho+ mtDNA and parents holding rho- mtDNA results in a moderately suppressive phenotype.

(1) What does entirely respiration competent progeny mean? We should not ignore moderately suppressive phenotype.

(2) Why can Pet127 protein affect more on wild-type mtDNA but not on HS rho- mtDNA?

Reviewer #2: PLOS Genetics D-21-00569

The authors of this manuscript have re-examined one of the oldest puzzles in yeast genetics: why to do some deleted versions of mtDNA, termed hypersuppressive (HS) rho- mtDNAs (or petites), eliminate complete rho+ mtDNAs when cells bearing these two forms are mated to produce diploids. Two competing, but not mutually exclusive, models have been considered: the HS mtDNAs might simply out-replicate the rho+ mtDNAs, or the HS mtDNAs might interact with and destroy the rho+ mtDNAs. The replicative interpretation is generally favored, and as a result the sequences found in HS mtDNAs have been termed ORI for origins of replication.

This paper demonstrates that a balance between the interacting proteins Pet127 (a mitochondrial RNA exonuclease) and Rpo41 (the mitochondrial RNA polymerase) is necessary for the full hypersuppressive effect, and that this appears to depend upon normal levels of transcription activity. The authors also present evidence that hypersuppressivity involves destruction of rho+ mtDNA, although this is problematic.

For this study the authors generated a collection of HS strains and showed that, consistent with previous studies, they contain only tandem repeats of three mtDNA sequences known as ORI2, ORI3 and ORI5. The bulk of their experiments were done with those containing ORI5.

The most interesting and novel of the results here flow from their discovery that overexpression of Pet127 reduces the hypersuppressivity of Ori5-1 HS mtDNA. Pet127 is a mitochondrial RNA exonuclease, and is known to physically interact with the mitochondrial RNA polymerase Rpo41 (which the authors confirm here). The authors show that Pet127 nuclease activity is not required for reduction of hypersuppressivity by mutating its active site. Instead it works through binding with the mtRNA polymerase Rpo41. Overexpression of both Pet127 and Rpo41 does not lead to reduction in hypersuppressivity. Thus, there is a balance between these proteins in wild-type: when it is disturbed by Pet127 overproduction hypersuppressivity is reduced. The experiments with truncated forms of Pet127 establish the domain required for this interaction.

The authors state that the effects of Pet127 overproduction are much less evident when tested against ORI2 and ORI3 containing HS strains. The text cites Fig. 3d for evidence, but that figure is missing from the paper (although its legend is there). It appears therefore, that it may not be possible to generalize about the mechanisms of hypersuppressivity.

Major points:

1) There are significant weaknesses in the interpretations of the experiment of Fig. 2a (this should be Fig 2 since there is no 2b). This experiment interrogates multiple diploid clones generated by mating HS-Ori5-1 with rho+ cells, for retention of various sequences mtDNA. These clones comprise populations of cells generated by events occurring during zygote formation and roughly 20 mitotic generations thereafter, and may each contain cells of multiple mitochondrial genotypes. It seems that some of these clones contain rho- mtDNAs that have retained large pieces of rho+ mtDNA. For example Fig. 2a shows that colony 5 lacks Cox2, Cox3 and Cox1-Cterm, but has ATP9, and Cox1-Nterm, while colony 8 retains far more Cox1-Nterm than Cox1-Cterm. (Since they are diploids, it is not possible to test whether these cells retain HS activity.) Thus, it is not surprising that these colonies are respiration incompetent, contrary to page 10, line 173. The authors state (Pg. 10 line 180) "From these data we conclude that rho+ mtDNA is lost in a piecemeal fashion rather than all at once." This implies that they interpret the results as a kinetic analysis. Instead, it is an endpoint analysis of processes that occurred in the zygotes and during the mitotic cell divisions that produced colonies from the zygotes. To generate a time-resolved picture of these events, one would have to follow the fates of mtDNAs during zygote formation and early cell divisions, perhaps using SMRT sequencing of bulk DNA or even single zygotes/cells.

2) The authors employ a temperature sensitive mutation in Rpo41 to test whether its activity affects hypersuppressivity of HS-ORI5. Activity was normal at permissive temperature (30°), but somewhat reduced at an intermediate temperature (34°). The experiment cannot be done at nonpermissive temperature since rho+ mtDNA is unstable in the absence of transcription. The authors conclude that Rpo41 activity is "required" for hypersuppressivity. This is an over-interpretation of the data. The extended discussion of how reduced Rpo41 activity could reduce the hypersuppressiveness of ORI5-containing mtDNAs is overly long and speculative.

3) Discussion pg. 22, line 442: The authors write: "The replicative advantage model fails to account for how the rho+ mtDNA vanishes from cells. This is problematic because a heteroplasmic cell having both rho+ and HS mtDNA should be able to respire, as the rho+ mtDNA should behave in a dominant way within a cell." It is not obvious that is true: the heteroplasmic state is extremely unstable in baker's yeast due to the observation of rapid mitotic segregation. This segregation, thought to be due to the seeding of buds with only a few molecules of mtDNA, leads rapidly to pure mitochondrial genotypes from heteroplasmic mother cells. Once the heteroplasmic mother cells become senescent due to replicative aging they will no longer grow on any medium. If the HS rho- mtDNA has a replicative advantage it would be preferentially passed on during mitotic segregation.

Minor points:

1) Reference 41 (Chambers & Gingold) does not list the year or journal.

2) Fig 2a What is the band amplified by the ATP9 primers that shows up in the rho0 control and is variable in the other lanes? (There is no Fig 2b, so this should just be Fig.2.)

3) As noted above, Fig. 3d is missing from the manuscript.

4) Fig. 4f: the host yeast strain(s) transformed with overexpressing plasmids in the experiment of Fig. 4f are not indicted in the strain table. Were these strains expressing wild-type PET127? If so, could that have played a role in assisting the truncated Pet127 protein interact with Rpo41?

5) Figs 1a and 1b (but not 1c) could be moved to the supplemental materials as they do not represent novel findings or contain information necessary to understand the subsequent text.

6) Does the absence of Pet127 in diploids have any effect on hypersuppressivity?

Reviewer #3: The manuscript by Corbi and Amon reports the role of PET127 in the biased segregation of a yeast hypersuppressive (HS) mitochondrial (mt) mutant genome (HS ORI5-1) when mated to the wild type rho+ mitochondrial genome. In an overexpression screen they found that high copy number plasmids containing PET127 increase the fraction of mitotic descendants from the heteroplasmid zygote that retain the rho+ genome. Work from others suggested that Pet127 is a mt exonuclease that is responsible for processing 5’ ends of mt RNAs and that it binds to the mtDNA RNA polymerase (encoded by RPO41). Corbi and Amon confirmed the physical interaction with Rpo41. They made an exonuclease-null mutation of Pet127 and showed that this mutation results in the accumulation of intergenic mtRNA sequences. They physically mapped the region of Pet127 that binds to Rpo41. In crosses with HS ORI5-1, overexpression of the nuclease null mutant but not the Rpo41 binding mutant reduced the inheritance of the HS ORI5-1 genome. They propose that it is Pet127’s inhibition of mtRNA transcription by Rpo41 that is responsible for the improved recovery of rho+ genomes from the heteroplasmic zygotes and support this hypothesis by co-overexpressing both PRO41 and PET127. In addition, they follow the fate of the mt rho+ genome in the descendants of the heteroplasmic zygotes by PCR and find that remnants of the rho+ genome can be detected in the descendants, suggesting that the HS ORI5-1 genome promotes physical destruction of the rho+ genome.

Suppressiveness refers to the propensity of a massively deleted version of the mt genome (a rho- genome that is amplified in head-to tail concatomers) to out-compete the rho+ genome in vegetative growth following mating. The degree of suppressiveness varies on the specific region and/or size of the region of the rho+ genome that is amplified. A few region of the rho+ genome give rise to the hypersuppressive phenotype, but only when the amplicon size is extremely small. These regions contain sequences that have been named rep or ori. It is one specific HS petite that was used to identify Pet127 in this study. The phenomenon of hypersuppressive mitochondrial petites has been around for a long time, but the mechanisms responsible are still unresolved, in part because different HS petites behave differently.

In this manuscript, the authors found an interaction between Rpo41 and Pet127 that influences the HS phenotype of one specific petite (HS ORI5-1) but also show that it clearly doesn’t have the same effect on inheritance of two other HS petites (ORI2-1 and ORI3-1). Moreover, it doesn’t affect HS ORI5-2’s inheritance, even though this larger rho- (with a different junction sequence) contains all of the sequences present in HS ORI5-1. Unfortunately, the authors tend to overgeneralize their findings and the reader is left with the misconception that they have uncovered a universal HS mechanism. I don’t believe it is deliberate, but it would help if the authors were more specific when making conclusions and also talked about some of these puzzling properties of HS petites.

A major concern

The last third of the paper describes transcript abundance data for various mutants and overexpression scenarios. What I find missing from these studies is an estimate of the template abundance. There could be difference is the number of copies per cell of the mtDNA templates that influences the abundance of transcripts but that doesn’t necessarily reflect RNA polymerase activity. The normalizations they perform are comparing ACT1 mRNA to ACT1 DNA and mtRNA to its mtDNA template. But no comparison of mtDNA to ACT1 DNA is made. The copy number of mtDNA can vary and be influenced by state of growth and mutant background. The gal-inducible experiments suffer less from this concern as the cells are only able to go through ~3 doublings (?) so there is less time for a copy number change, but even so, copy number should be checked. I have the same concerns about the ts-RPO41strain grown at the semi-permissive temperature.

Other concerns:

1) The authors frequently draw attention to changes in measurements that don’t reach significance. That puzzles me and I find it misleading.

2) The authors need to be more precise when summarizing their results as their findings specifically affect only one HS petite genome and not the other three they isolated.

3) Describing the HS regions of the rho+ genome as replication origins is an overstatement. People have proposed that they are origins but I do not know of any study that shows that replication actually initiates at these sites. Rolling circle replication started by invasion of a DNA end is a model with considerable support.

4) The finding that regions of the rho+ genome persists is interesting. The only evidence is from PCR which is notoriously non-quantitative. Because so little attention was paid to the specifics of the PCR, I am wondering whether adequate controls were performed to be sure that differences found between primer pairs wasn’t just due to the fact that different primer pairs were used. Did the authors spike in different amounts of the rho+ parent into the DNA from the haploid rho- to convince themselves (and the readers) that the absence of product was biologically real? If this observation is real, it would be worth pursuing by Southern blotting or PacBio sequencing to look for the novel junction sequences.

**Have all data underlying the figures and results presented in the manuscript been provided?**

Reviewer #1: Yes

Reviewer #2: Yes

Reviewer #3: Yes

PLOS authors have the option to publish the peer review history of their article (what does this mean?). If published, this will include your full peer review and any attached files.

Reviewer #1: **Yes: **Feng LING

Reviewer #2: No

Reviewer #3: No

---

## [Decision Letter · Decision Letter 1]

7 Sep 2021

Dear Dr Corbi,

We are pleased to inform you that your manuscript entitled "Decreasing Mitochondrial RNA Polymerase Activity Reverses Biased Inheritance of Hypersuppressive mtDNA" has been editorially accepted for publication in PLOS Genetics. Congratulations!

Yours sincerely,

David M. MacAlpine

Guest Editor

PLOS Genetics

Gregory P. Copenhaver

Editor-in-Chief

PLOS Genetics

Comments from the reviewers (if applicable):

Reviewer's Responses to Questions

**Comments to the Authors:**

Reviewer #2: This revision has addressed a number of points raised, with respect to clearer statements about conclusions, stain identification, and clarification of some experimental details. The abstracts contains an addition that clarifies somewhat the arguement that hypersupressivity may involve active destruction of rho+ mtDNA. The authors also provide some supporting data in their rebuttal (but not the paper itself), that confirms published results of previous workers. I am more convince now about the strength of their interpretations, and they are more appropriately described.

**Have all data underlying the figures and results presented in the manuscript been provided?**

Reviewer #2: Yes

PLOS authors have the option to publish the peer review history of their article (what does this mean?). If published, this will include your full peer review and any attached files.

Reviewer #2: No

**Data Deposition**

http://datadryad.org/submit?journalID=pgenetics&manu=PGENETICS-D-21-00569R1

**Press Queries**

---

## [Editor Report · Acceptance letter]

14 Oct 2021

PGENETICS-D-21-00569R1 

Decreasing Mitochondrial RNA Polymerase Activity Reverses Biased Inheritance of Hypersuppressive mtDNA 

Dear Dr Corbi, 

We are pleased to inform you that your manuscript entitled "Decreasing Mitochondrial RNA Polymerase Activity Reverses Biased Inheritance of Hypersuppressive mtDNA" has been formally accepted for publication in PLOS Genetics! Your manuscript is now with our production department and you will be notified of the publication date in due course.

With kind regards,

Andrea Szabo

PLOS Genetics

On behalf of:
